# Provable Acceleration of Nesterov's Accelerated Gradient for Rectangular Matrix Factorization and Linear Neural Networks

**Zhenghao Xu**
Georgia Institute of Technology
zhenghaoxu@gatech.edu

**Yuqing Wang**
University of California, Berkeley
yq.wang@berkeley.edu

**Tuo Zhao**
Georgia Institute of Technology
tourzhao@gatech.edu

**Rachel Ward**
University of Texas at Austin
rward@math.utexas.edu

**Molei Tao**
Georgia Institute of Technology
mtao@gatech.edu

## Abstract

We study the convergence rate of first-order methods for rectangular matrix factorization, which is a canonical nonconvex optimization problem. Specifically, given a rank-$r$ matrix $\mathbf{A} \in \mathbb{R}^{m \times n}$, we prove that gradient descent (GD) can find a pair of $\epsilon$-optimal solutions $\mathbf{X}_T \in \mathbb{R}^{m \times d}$ and $\mathbf{Y}_T \in \mathbb{R}^{n \times d}$, where $d \geq r$, satisfying $\|\mathbf{X}_T \mathbf{Y}_T^\top - \mathbf{A}\|_\mathrm{F} \leq \epsilon \|\mathbf{A}\|_\mathrm{F}$ in $T = O(\kappa^2 \log \frac{1}{\epsilon})$ iterations with high probability, where $\kappa$ denotes the condition number of $\mathbf{A}$. Furthermore, we prove that Nesterov's accelerated gradient (NAG) attains an iteration complexity of $O(\kappa \log \frac{1}{\epsilon})$, which is the best-known bound of first-order methods for rectangular matrix factorization. Different from small balanced random initialization in the existing literature, we adopt an unbalanced initialization, where $\mathbf{X}_0$ is large and $\mathbf{Y}_0$ is $0$. Moreover, our initialization and analysis can be further extended to linear neural networks, where we prove that NAG can also attain an accelerated linear convergence rate. In particular, we only require the width of the network to be greater than or equal to the rank of the output label matrix. In contrast, previous results achieving the same rate require excessive widths that additionally depend on the condition number and the rank of the input data matrix.

## 1   Introduction

Nonconvex optimization is pervasive in the training of modern machine learning models. Despite the success of first-order methods in practice, theoretical understanding of their convergence properties is limited even for simple nonconvex problems. Take the rectangular low-rank matrix factorization problem as an example, which is a canonical nonconvex problem:

$$\min_{\mathbf{X} \in \mathbb{R}^{m \times d}, \mathbf{Y} \in \mathbb{R}^{n \times d}} f(\mathbf{X}, \mathbf{Y}) = \frac{1}{2} \left\| \mathbf{A} - \mathbf{X}\mathbf{Y}^\top \right\|_\mathrm{F}^2, \tag{1}$$

where we solve for two small matrices $\mathbf{X} \in \mathbb{R}^{m \times d}$ and $\mathbf{Y} \in \mathbb{R}^{n \times d}$ to approximate a big rank-$r$ target matrix $\mathbf{A} \in \mathbb{R}^{m \times n}$ with $r \ll \min(m, n)$ and $m, n$ not necessarily equal. Specifically, we consider the over-parameterized regime where $d \geq r$, so that the global minimum of (1) is zero. While various

38th Conference on Neural Information Processing Systems (NeurIPS 2024).

direct methods exist for solving (1), we focus on understanding the global convergence behaviors of first-order methods applied to such a nonconvex problem, with the motivation of gathering insight into the training dynamics of neural networks.

Most existing results study the simplest first-order method, gradient descent (GD), under different initialization schemes. Note that the initialization scheme matters to convergence analysis[1], due to the fact that (1) is a nonconvex and nonsmooth[2] optimization problem. Thus, proper initialization is important for the fast convergence rates of first-order methods. Ye and Du [2021] show that with small Gaussian random initialization, GD can find $\mathbf{X}_T$ and $\mathbf{Y}_T$ such that $f(\mathbf{X}_T, \mathbf{Y}_T) \leq \epsilon$ in $T = O(d^4(m+n)^2\kappa^4 \log \frac{1}{\epsilon})$ iterations with high probability, where $\kappa$ denotes the condition number. Jiang et al. [2023] improve this result to $O(\kappa^3 \log \frac{1}{\epsilon})$ which has no explicit dimensional dependence on $m$ and $n$. These analyses rely on balanced initialization where entries of $\mathbf{X}_0$ and $\mathbf{Y}_0$ have the same variance so that the iterates are guaranteed to stay in a smooth region.

Moreover, we remark that to the best of our knowledge, we are not aware of any existing theoretical results on rectangular matrix factorization analyzing the global convergence rate of more advanced first-order methods such as Nesterov's accelerated gradient (NAG), which has been proved to achieve faster rates for smooth convex optimization problems [Nesterov, 2013].

Recently, Ward and Kolda [2023] showed that by using an unbalanced random initialization where $\mathbf{X}_0$ is larger than $\mathbf{Y}_0$, alternating gradient descent (AltGD) that alternatingly optimizes $\mathbf{X}_t$ and $\mathbf{Y}_t$ via gradient steps can achieve $O(d^2(d-r+1)^{-2}\kappa^2 \log \frac{1}{\epsilon})$ iteration complexity. However, their analysis is specifically designed for AltGD and not applicable to GD, let alone more advanced methods such as NAG which are nevertheless widely used in machine learning practice. Two questions naturally arise here:

Q1: *Can GD achieve the same convergence rate as AltGD for* (1)*?*

Q2: *Can more advanced first-order methods (e.g., NAG) achieve faster convergence rate for* (1)*?*

● **Main Results**. We answer the two questions above affirmatively by developing a new theory on first-order methods for (1). Specifically, we consider an unbalanced initialization scheme $\mathbf{X}_0 = c\mathbf{A}\mathbf{\Phi}$ and $\mathbf{Y}_0 = 0$, where $c > 0$ is a large constant and $\mathbf{\Phi}$ is a Gaussian random matrix. Note that our initialization of $\mathbf{X}_0$ is the same as that in Ward and Kolda [2023], but they initialized $\mathbf{Y}_0$ using a small Gaussian random matrix. This modification is mainly for simpler analysis and makes little difference in practice. Under our new initialization scheme, we first prove an $O(d^2(d-r+1)^{-2}\kappa^2 \log \frac{1}{\epsilon})$ iteration complexity for GD (Theorem 1), matching that of AltGD in Ward and Kolda [2023]. Our analysis is based on a new theoretical framework different from Ward and Kolda [2023] and can be further extended to analyzing NAG. We then show that NAG can attain a provable acceleration with an $O(d(d-r+1)^{-1}\kappa \log \frac{1}{\epsilon})$ iteration complexity (Theorem 2). We discuss the tightness of our results (Remark 2) and conduct numerical experiments for validation (Section 5). Empirically, we observe that NAG exhibits a much faster rate than GD and our bounds are quite tight.

Our analysis technique can also be applied to linear neural networks. We consider unbalanced initialization similar to the one for (1). We show that NAG can achieve an accelerated convergence rate for each overparameterization level (Corollaries 1 to 3), under the commonly adopted interpolation assumption (Assumption 1, see e.g. Du and Hu 2019). In particular, we only require the network width to be greater than the rank of the output matrix.

● **Additional Related Work**. For matrix factorization, there is a large body of works focusing on the *symmetric* case, where $\mathbf{A}$ is positive semidefinite and $\mathbf{A} = \mathbf{X}\mathbf{X}^\top$ [Bhojanapalli et al., 2016, Li et al., 2018]. However, these analyses are difficult to generalize to the rectangular case (1) due to the additional unbalanced scaling issue[3]. To overcome this, additional *balancing regularization* is often required [Tu et al., 2016, Park et al., 2017, Zhang et al., 2021, Bi et al., 2022], which changes the objective function in (1). Du et al. [2018] show that GD can automatically balance the two factors hence explicit regularization is not necessary, but they only establish linear convergence rate for rank-1 matrix and cannot generalize to rank-$r$ case. Some other works remove this regularization for the

---

[1]There are some works [Wang et al., 2022, 2023] proving convergence of GD for general initialization under large learning rate and similar objective functions, but nonasymptotic convergence analysis is very challenging and highly dependent on initialization.

[2]Here, the nonsmoothness refers to the lack of uniform Lipschitz constant for the gradient in the full domain.

[3]In the symmetric case, the solution's uniqueness is up to rotation, whereas in (1) it is also up to scaling.

general matrix sensing problem and show linear convergence rate for general ranks [Ma et al., 2021, Tong et al., 2021a,b]. These results do not directly apply to our setting as they require singular value decomposition (SVD) at initialization, which consumes roughly the same amount of computation as solving (1). Moreover, these works only consider *exact parameterization* ($d = r$), leaving out the overparameterization regime ($d > r$). Overparameterization may heavily slow down convergence due to the possible singularity of iterates [Stöger and Soltanolkotabi, 2021], thus some works consider preconditioning for acceleration [Zhang et al., 2023, Xu et al., 2023]. These preconditioned methods are specifically tailored to symmetric factorization and are not directly comparable with the first-order methods we consider, as their algorithms not only use the gradient. For algorithmic acceleration, Zhou et al. [2020] first propose a computationally tractable modified Nesterov's method for general loss function $f$ that is $L$-smooth $\mu$-strongly convex to the product $\mathbf{X}\mathbf{Y}^\top$. However, their method still requires balancing regularization when applied to rectangular matrices, and SVD-based initialization that dominates the computation. Moreover, their acceleration pertains to the condition number $L/\mu$ of the loss function rather than $\kappa$ of the target matrix, on which their dependence is $O(\kappa^2)$.

For linear neural networks, Du and Hu [2019] and Hu et al. [2020] show linear convergence of GD with Gaussian and orthogonal initialization respectively, and Min et al. [2021] studies convergence rate of gradient flow (GF) for unbalanced initialization. Wang et al. [2021] show that Polyak's heavy ball (HB) method [Polyak, 1964] attains accelerated convergence rate with orthogonal initialization. Liu et al. [2022] further investigate NAG and show a similar accelerated rate for Gaussian initialization. All these previous works consider sufficiently wide networks that depend on the output dimension, the rank, and the condition number of input. The results are summarized in Table 1.

Table 1: Results for linear neural networks. All results in table are based on the assumption $\mathbf{L} = \mathbf{A}\mathbf{D}$ for some $\mathbf{A}$ with $\mathrm{cond}(\mathbf{A}) = O(1)$, where $\mathbf{D}$ denotes the input data, $\mathbf{L}$ denotes the output data, $d_{\mathrm{out}}$ denotes the output dimension, $\delta$ denote the failure probability, $r = \mathrm{rank}(\mathbf{D})$, $\bar{r} = \mathrm{rank}(\mathbf{L})$, $\tilde{r} = \|\mathbf{D}\|_{\mathrm{F}}^2 / \|\mathbf{D}\|^2$, $\kappa = \mathrm{cond}^2(\mathbf{D})$, $\kappa_1 = O(\kappa^2)$, $\kappa_2 = O(\kappa)$.

| Algorithm | Initialization | Width | Rate |
|---|---|---|---|
| GD [Du and Hu, 2019] | Gaussian | $\Omega\left(r\kappa^3(d_{\mathrm{out}} + \log\frac{r}{\delta})\right)$ | $(1 - \frac{3}{4\kappa})^t$ |
| GD [Hu et al., 2020] | Orthogonal | $\Omega\left(\tilde{r}\kappa^2(d_{\mathrm{out}} + \log\frac{r}{\delta})\right)$ | $(1 - \frac{1}{4\kappa})^t$ |
| HB [Wang et al., 2021] | Orthogonal | $\Omega\left(\frac{\kappa^5}{\|\mathbf{D}\|^2}(d_{\mathrm{out}} + \log\frac{r}{\delta})\right)$ | $(1 - \frac{1}{4\sqrt{\kappa}})^t$ |
| NAG [Liu et al., 2022] | Gaussian | $\Omega\left(r\kappa^5(d_{\mathrm{out}} + \log\frac{r}{\delta})\right)$ | $(1 - \frac{1}{2\sqrt{\kappa}})^t$ |
| NAG (ours, Corollary 1) | Unbalanced (12) | $\geq \bar{r} + \Omega(\log\frac{1}{\delta})$ | $(1 - \frac{1}{2\sqrt{\kappa_1}})^t$ |
| NAG (ours, Corollary 2) | Unbalanced+Orth (13) | $\geq \bar{r}$ | $(1 - \frac{1}{2\sqrt{\kappa}})^t$ |
| NAG (ours, Corollary 3) | Unbalanced (14) | $\geq d_{\mathrm{out}} + \Omega(\log\frac{1}{\delta})$ | $(1 - \frac{1}{2\sqrt{\kappa_2}})^t$ |

• **Notations**. Throughout this paper, $\|\cdot\|$ denotes the Euclidean norm of a vector or the spectral norm of a matrix, and $\|\cdot\|_{\mathrm{F}}$ denotes the Frobenius norm of a matrix. For any matrix, $\sigma_i(\cdot)$ denotes its $i$-th largest singular value. For a square matrix, $\lambda_i(\cdot)$ denotes its $i$-th largest eigenvalue. For a nonzero positive semidefinite matrix, $\lambda_{\max}(\cdot)$ and $\lambda_{\min}(\cdot)$ denote its largest and smallest nonzero eigenvalues respectively. For a matrix $\mathbf{X}$, we use $\mathrm{col}(\mathbf{X})$ to denote its column space, $\ker(\mathbf{X})$ to denote its kernel space and define $\mathrm{cond}(\mathbf{X}) \coloneqq \|\mathbf{X}\|\,\|\mathbf{X}^\dagger\|$ as its condition number, where $\mathbf{X}^\dagger$ denotes the pseudoinverse of $\mathbf{X}$. For any positive integer $n$, $\mathbf{I}_n$ denotes the identity matrix of size $n$. We use $\otimes$ to denote the Kronecker product between matrices, $\oplus$ to denote the direct sum of vector spaces, and $\mathrm{vec}(\cdot)$ to denote the column-first vectorization of a matrix. We use $\mathcal{N}(\mu, \sigma^2)$ to denote Gaussian distribution with mean $\mu$ and variance $\sigma^2$.

## 2 Results for Matrix Factorization

We start with formalizing our initialization scheme for matrix factorization problem (1). Let $\mathbf{\Phi} \in \mathbb{R}^{n \times d}$ be a Gaussian random matrix with i.i.d. entries $[\mathbf{\Phi}]_{i,j} \sim \mathcal{N}(0, 1/d)$. We initialize

$$\mathbf{X}_0 = c\mathbf{A}\mathbf{\Phi}, \quad \mathbf{Y}_0 = 0, \tag{2}$$

where $c > 0$ is a constant to be specified later. Typically, we require $c$ to be larger than a certain threshold, which depends on the dimensions, the extreme singular values of $\mathbf{A}$, and possibly the

condition number of $\mathbf{X}_0$. We note that changing $c$ would not affect $\mathrm{cond}(\mathbf{X}_0)$, hence there is no recursive definition. As we mentioned, (2) is a modified version of the initialization in Ward and Kolda [2023], where we replace the small random Gaussian matrix $\mathbf{Y}_0$ by $0$ and choose $c$ independently of the step size. We set $\mathbf{Y}_0 = 0$ mainly for simplicity, and our analysis can be extended to the case where $\mathbf{Y}_0$ is a sufficiently small Gaussian random matrix. While the initialization of $\mathbf{X}_0$ differs from standard Gaussian initialization, it has the following interpretation: Suppose we start from $t = -1$ and let $\mathbf{X}_{-1} = c' \mathbf{\Phi}'$ and $\mathbf{Y}_{-1} = c'' \mathbf{\Phi}$ for some $0 < c' \ll c'' \ll 1$ and Gaussian random matrix $\mathbf{\Phi}'$, then by taking a gradient step with step size $c/c''$ we get $\mathbf{X}_0 \approx c\mathbf{A}\mathbf{\Phi}$ and $\mathbf{Y}_0 \approx 0$. This initialization of $\mathbf{X}_0$ also coincides with the first step of randomized singular value decomposition, which is also referred to as sketching (see e.g. [Halko et al., 2011]).

## 2.1  Gradient Descent

With initialization (1), we can analyze the global convergence rates of various first-order methods. Consider gradient descent (GD) first. The gradient of the squared Frobenius error in (1) is given by

$$\nabla_X f(\mathbf{X}, \mathbf{Y}) = (\mathbf{X}\mathbf{Y}^\top - \mathbf{A})\mathbf{Y}, \quad \nabla_Y f(\mathbf{X}, \mathbf{Y}) = (\mathbf{X}\mathbf{Y}^\top - \mathbf{A})^\top \mathbf{X}.$$

For $t \geq 0$, the GD update with constant step size $\eta > 0$ is written as

$$\begin{pmatrix} \mathbf{X}_{t+1} \\ \mathbf{Y}_{t+1} \end{pmatrix} = \begin{pmatrix} \mathbf{X}_t - \eta(\mathbf{X}_t \mathbf{Y}_t^\top - \mathbf{A})\mathbf{Y}_t \\ \mathbf{Y}_t - \eta(\mathbf{X}_t \mathbf{Y}_t^\top - \mathbf{A})^\top \mathbf{X}_t \end{pmatrix}. \tag{3}$$

Let $\mathbf{R}_t := \mathbf{X}_t \mathbf{Y}_t^\top - \mathbf{A}$ denote the residual, then $f(\mathbf{X}_t, \mathbf{Y}_t) = \frac{1}{2} \|\mathbf{R}_t\|_{\mathrm{F}}^2$. We have the following convergence rate for GD.

**Theorem 1** (GD convergence rate). *For $0 < \tau < c_1$, denote $\delta = 3e^{-(d-r+1)\cdot\min\{\log\frac{1}{c_1\tau}, c_2, \frac{1}{2}\}}$, where $c_1$ and $c_2$ are universal constants. Denote $L = \sigma_1^2(\mathbf{X}_0)$, $\mu = \sigma_r^2(\mathbf{X}_0)$. Let $\eta = \frac{2}{L+\mu}$, $c \geq \underline{c} := \frac{\sqrt{d}\sigma_r(\mathbf{A})}{12\tau(\sqrt{d}-\sqrt{r-1})} \sqrt{\frac{\mathrm{cond}^4(\mathbf{X}_0)\|\mathbf{A}\|_{\mathrm{F}}}{\mathrm{cond}^2(\mathbf{X}_0)-1}}$ be a sufficiently large constant. Then with $c$ plugged in initialization (2), GD returns $\mathbf{X}_t$ and $\mathbf{Y}_t$ with probability at least $1 - \delta$ such that*

$$\|\mathbf{R}_t\|_{\mathrm{F}} \leq \frac{3c^2\sigma_1^2(\mathbf{A})}{64}\left(1 - \frac{\mu}{L}\right)^t.$$

*In particular, if $c = \underline{c}$, then GD finds $\|\mathbf{R}_T\|_{\mathrm{F}} \leq \epsilon \|\mathbf{A}\|_{\mathrm{F}}$ in*

$$T = O\left(\frac{d^2\kappa^2}{\tau^2(d-r+1)^2} \cdot \log\frac{C}{\epsilon}\right)$$

*iterations, where $C = \frac{27\tau^2(d-r+1)^2}{16d^2}\frac{\mathrm{cond}^4(\mathbf{X}_0)\kappa^2}{\mathrm{cond}^2(\mathbf{X}_0)-1}$.*

Theorem 1 shows that GD converges in $O(d^2(d-r+1)^{-2}\kappa^2\log\frac{1}{\epsilon})$ iterations with initialization (2), and the constant prefactor does not have dependence on the ambient dimension $m$ and $n$. This matches the convergence rate for AltGD derived in Ward and Kolda [2023]. The step size $\frac{2}{L+\mu}$ is commonly used in optimization literature and leads to optimal convergence rate [Nesterov, 2013]. While the bound on $\|\mathbf{R}_t\|_{\mathrm{F}}$ in Theorem 1 does not explicitly depend on $\|\mathbf{A}\|_{\mathrm{F}}$, the norm still affects the convergence rate through the choice of $c$ defined in (2). When $c = \underline{c} = O(\sqrt{\|\mathbf{A}\|_{\mathrm{F}}})$, the bound linearly depends on $\|\mathbf{A}\|_{\mathrm{F}}$. When $c > \underline{c}$, the bound linearly depends on $c^2$, which dominates $\|\mathbf{A}\|_{\mathrm{F}}$.

## 2.2  Nesterov's Accelerated Gradient

We then consider Nesterov's accelerated gradient (NAG) method [Nesterov, 2013] applied to (1). We take the form of NAG that is originally designed for smooth strongly convex loss function $\ell$:

$$z_{t+1} = \tilde{z}_t - \eta\nabla\ell(\tilde{z}_t), \quad \tilde{z}_{t+1} = z_{t+1} + \beta(z_{t+1} - z_t),$$

where $\eta$ is the step size, $\beta$ is the momentum parameter, and $z$ or $\tilde{z}$ in our case consists of both $\mathbf{X}$ and $\mathbf{Y}$. If we focus on the $\{\tilde{z}_t\}$ sequence with $\tilde{z}_t = (\mathbf{X}_t, \mathbf{Y}_t)$ and plug in the objective function in (1), then with $\mathbf{X}_{-1} = \mathbf{X}_0$ and $\mathbf{Y}_{-1} = \mathbf{Y}_0$, the NAG update is given by

$$\begin{pmatrix} \mathbf{X}_{t+1} \\ \mathbf{Y}_{t+1} \end{pmatrix} = \begin{pmatrix} (1+\beta)(\mathbf{X}_t - \eta\mathbf{R}_t\mathbf{Y}_t) - \beta(\mathbf{X}_{t-1} - \eta\mathbf{R}_{t-1}\mathbf{Y}_{t-1}) \\ (1+\beta)(\mathbf{Y}_t - \eta\mathbf{R}_t^\top\mathbf{X}_t) - \beta(\mathbf{Y}_{t-1} - \eta\mathbf{R}_{t-1}^\top\mathbf{X}_{t-1}) \end{pmatrix}. \tag{4}$$

We have the following convergence rate for NAG.

**Theorem 2** (NAG convergence rate). *For $0 < \tau < c_1$, define $\delta$ as in Theorem 1. Denote $L = \sigma_1^2(\mathbf{X}_0)$, $\mu = \sigma_r^2(\mathbf{X}_0)$. Let $\eta = \frac{1}{L}$, $\beta = \frac{\sqrt{L}-\sqrt{\mu}}{\sqrt{L}+\sqrt{\mu}}$, $c \geq \underline{c} := 29\sqrt{\frac{d(2\sqrt{d}+\sqrt{r})\|\mathbf{A}\|_{\mathrm{F}}\cdot\kappa}{\tau^3(\sqrt{d}-\sqrt{r-1})^3\sigma_r^2(\mathbf{A})}}$ be a constant. Then with $c$ plugged in initialization (2), NAG returns $\mathbf{X}_t$ and $\mathbf{Y}_t$ with probability at least $1 - \delta$ such that*

$$\|\mathbf{R}_t\|_{\mathrm{F}} \leq \frac{c^2\sigma_1^2(\mathbf{A})}{64\,\mathrm{cond}(\mathbf{X}_0)}\left(1 - \frac{\sqrt{\mu}}{2\sqrt{L}}\right)^t.$$

*In particular, if $c = \underline{c}$ then NAG finds $\|\mathbf{R}_T\|_{\mathrm{F}} \leq \epsilon\|\mathbf{A}\|_{\mathrm{F}}$ in*

$$T = O\left(\frac{d\kappa}{\tau(d-r+1)}\cdot\log\frac{C}{\epsilon}\right)$$

*iterations, where $C = \frac{841d(2\sqrt{d}+\sqrt{r})}{64\tau^3(\sqrt{d}-\sqrt{r-1})^3}\cdot\frac{\kappa^3}{\mathrm{cond}(\mathbf{X}_0)}$.*

Theorem 2 shows that NAG can achieve $O(d(d-r+1)^{-1}\kappa\log\frac{1}{\epsilon})$ iteration complexity with high probability. The dependence on the condition number $\kappa$ is improved from being quadratic to linear. Moreover, the dependence on the dimension is also improved. As shown in Theorem 1, the GD iteration number has an $O(d^2)$ dependence in the worst case ($d = r$). Here, NAG has at most $O(d)$ dependence. The level of overparameterization $d$ will affect both the convergence rate and the probability of success. To ensure a small fail probability $\delta$, it requires $d = r - 1 + \Omega(\log\frac{1}{\delta})$. Again, the step size $\frac{1}{L}$ and momentum $\frac{\sqrt{L}-\sqrt{\mu}}{\sqrt{L}+\sqrt{\mu}}$ are commonly used in the literature [Nesterov, 2013].

## 3 Proof Sketch for Convergence Rates

We now provide the proof sketch for Theorems 1 and 2. Our proof is based on induction. We start with the assumptions that $\mathbf{X}_t$ and $\mathbf{Y}_t$ are not too far from $\mathbf{X}_0$ and $\mathbf{Y}_0$ respectively and the initial residual is bounded by some constant, which are guaranteed at time $t = 0$. Given the induction assumptions, we then track the dynamics of residual $\mathbf{R}_t$ and decompose it into linear and higher-order parts. We can show that the linear part is contracted and the higher-order part shrinks exponentially, together implying that $\|\mathbf{R}_{t+1}\|_{\mathrm{F}} = O(\theta^t)$ for some $\theta \in (0, 1)$ and $\mathbf{X}_{t+1}$ and $\mathbf{Y}_{t+1}$ is still within a bounded region around initialization. This shows the induction assumptions for the next iterate, thus by invoking the induction we complete the proof.

The key to our proof is to show the contraction and its rate. Firstly, the linear part of the dynamics is not a contraction over the whole space, thus we need to identify in which subspace it is a contraction. Secondly, we need to quantify the rate of contraction to get global convergence rates. These necessitate the following proposition about the properties of $\mathbf{X}_0$ with initialization (2).

**Proposition 1.** *For any $\tau, c > 0$, $\mathbf{A} \in \mathbb{R}^{m\times n}$ being a rank-$r$ matrix with condition number $\kappa := \mathrm{cond}(\mathbf{A})$, $\mathbf{\Phi} \in \mathbb{R}^{n\times d}$ being a random matrix with i.i.d. entries from $\mathcal{N}(0, 1/d)$, the following holds for $\mathbf{X}_0 = c\mathbf{A}\mathbf{\Phi}$ with probability at least $1 - \delta$:*

$$\frac{\tau(\sqrt{d}-\sqrt{r-1})}{\sqrt{d}}c\cdot\sigma_r(\mathbf{A}) \leq \sigma_r(\mathbf{X}_0) \leq \sigma_1(\mathbf{X}_0) \leq \frac{2\sqrt{d}+\sqrt{r}}{\sqrt{d}}c\cdot\sigma_1(\mathbf{A}),$$

*where $\delta = 3e^{-\min\{(d-r+1)\log\frac{1}{c_1\tau}, c_2d, \frac{d}{2}\}}$, $c_1$ and $c_2$ are universal constants. When it holds, the condition number of $\mathbf{X}_0$ is bounded:*

$$\mathrm{cond}(\mathbf{X}_0) \leq \frac{2\sqrt{d}+\sqrt{r}}{\tau(\sqrt{d}-\sqrt{r-1})}\cdot\kappa \leq \frac{6d}{\tau(d-r+1)}\cdot\kappa.$$

By Proposition 1, the top singular value of $\mathbf{X}_0$ is bounded from above by $\sigma_1(\mathbf{A})$, and the $r$-th singular value of $\mathbf{X}_0$ is bounded from below by $\sigma_r(\mathbf{A})$, hence we have $\mathrm{cond}(\mathbf{X}_0) = O(\kappa)$. Moreover, $\mathbf{X}_0$ has rank $r$ with probability 1 and thus it preserves the column space of $\mathbf{A}$, i.e., $\mathrm{col}(\mathbf{X}_0) = \mathrm{col}(\mathbf{A})$. This subspace preservation property will be passed to subsequent iterations of first-order methods and is critical to our analysis. In particular, we will show this space corresponds to the contraction subspace.

### 3.1 Proof Sketch for GD Convergence Rate (Theorem 1)

As mentioned, we track the dynamics of $\mathbf{R}_t$ for GD to prove Theorem 1. Let $\mathbf{r}_t = \text{vec}(\mathbf{R}_t)$ denote the vectorized residual, then the GD update (3) corresponds to the following dynamics:

**Proposition 2** (GD dynamics). *Let $\mathbf{P}_t = \mathbf{X}_{t+1} - \mathbf{X}_t$ and $\mathbf{Q}_t = \mathbf{Y}_{t+1} - \mathbf{Y}_t$ denote the update steps for $t \geq 0$. Then GD (3) admits the following dynamics:*

$$\mathbf{r}_{t+1} = (\mathbf{I}_{mn} - \eta\mathbf{H}_0)\mathbf{r}_t + \boldsymbol{\xi}_t, \tag{5}$$

*where $\mathbf{H}_t = (\mathbf{Y}_t\mathbf{Y}_t^\top) \otimes \mathbf{I}_m + \mathbf{I}_n \otimes (\mathbf{X}_t\mathbf{X}_t^\top)$ and $\boldsymbol{\xi}_t = \eta(\mathbf{H}_0 - \mathbf{H}_t)\mathbf{r}_t + \text{vec}(\mathbf{P}_t\mathbf{Q}_t^\top)$.*

The linear part at time $t$ is $(\mathbf{I}_{mn} - \eta\mathbf{H}_t)\mathbf{r}_t$, which is approximately $(\mathbf{I}_{mn} - \eta\mathbf{H}_0)\mathbf{r}_t$ when $\mathbf{X}_t$ and $\mathbf{Y}_t$ are close to their initialization. The approximation error along with the higher-order term $\text{vec}(\mathbf{P}_t\mathbf{Q}_t^\top)$ is contained in $\boldsymbol{\xi}_t$. It follows immediately from Proposition 2 that

$$\mathbf{r}_{t+1} = (\mathbf{I}_{mn} - \eta\mathbf{H}_0)^{t+1}\mathbf{r}_0 + \sum_{s=0}^{t}(\mathbf{I}_{mn} - \eta\mathbf{H}_0)^{t-s}\boldsymbol{\xi}_s.$$

If $\mathbf{T}_{\text{GD}} := \mathbf{I}_{mn} - \eta\mathbf{H}_0$ is a contraction map, i.e., it has all eigenvalues bounded $|\lambda_i(\mathbf{T}_{\text{GD}})| \leq \rho$ for some $\rho \in [0, 1)$, and the nonlinear error $\boldsymbol{\xi}_t$ shrinks exponentially at rate $\theta \in (\rho, 1)$, then we have $\|\mathbf{r}_t\| = O(\theta^t)$. However, for $d < \min(m, n)/2$, $\mathbf{T}_{\text{GD}}$ cannot be a contraction map for any $\eta$, as the rank of $\mathbf{H}_0$ is at most $(m + n)d < mn$. In fact, if $\mathbf{X}_0$ is initialized as in (2), then $\text{rank}(\mathbf{H}_0) = nr < mn$ regardless of the choice of $d$. As $\mathbf{H}_0$ has no full rank, $\mathbf{T}_{\text{GD}}$ must have a non-trivial eigensubspace corresponding to eigenvalue 1. In the following lemma, we show that $\mathbf{r}_t$ and $\boldsymbol{\xi}_t$ are not in this "bad" subspace but rather in a contracted subspace as desired.

**Lemma 1** (Eigensubspace). *Let $\mathcal{H} \subseteq \mathbb{R}^{mn}$ denote the linear subspace containing all eigenvectors of $\mathbf{H}_0$ with positive eigenvalues. If $\mathbf{X}_0$ is initialized as in (2), then we have*

$$\mathcal{H} = (\text{col}(\mathbf{A}))^n \quad and \quad \{\mathbf{r}_t, \boldsymbol{\xi}_t\}_{t \geq 0} \subset \mathcal{H},$$

*where $\mathbf{H}_0$, $\mathbf{r}_t$ and $\boldsymbol{\xi}_t$ are defined as in Proposition 2.*

Given that $\mathbf{r}_t$ and $\boldsymbol{\xi}_t$ are in the contracted subspace $\mathcal{H}$ throughout all iterations, the convergence rate is determined by the contractivity of $\mathbf{T}_{\text{GD}}$ over this subspace, which corresponds to the condition number of $\mathbf{X}_0$ with initialization (2).

**Lemma 2** (GD contractivity). *Let $L = \sigma_1^2(\mathbf{X}_0)$, $\mu = \sigma_r^2(\mathbf{X}_0)$, and $\mathcal{H}$ be defined as in Lemma 1. Let $\eta \in (0, \frac{2}{L})$, then for any $\mathbf{v} \in \mathcal{H}$,*

$$\|\mathbf{T}_{\text{GD}}\mathbf{v}\| \leq \max\{|1 - \eta L|, |1 - \eta\mu|\}\|\mathbf{v}\|.$$

*In particular, if $\eta = \frac{2}{L+\mu}$, then $\|\mathbf{T}_{\text{GD}}\mathbf{v}\| \leq \frac{L-\mu}{L+\mu}\|\mathbf{v}\|$.*

By Lemmas 1 and 2, the linear part of GD dynamics contracts $\mathbf{r}_t$ and $\boldsymbol{\xi}_t$, and the rate of contraction is $\rho = \max\{|1 - \eta L|, |1 - \eta\mu|\}$. To complete the proof, it remains to bound the magnitude of error $\boldsymbol{\xi}_t$ and show induction conditions for the next iteration. This is guaranteed by the following lemma.

**Lemma 3** (Nonlinear error). *If there exist $\theta \in (0, 1)$ and some constants $C_1$ and $C_2$ such that for any $s \leq t$, the GD dynamics (5) yields $\|\mathbf{r}_s\| \leq C_1\theta^s\|\mathbf{r}_0\|$, $\|\mathbf{X}_s - \mathbf{X}_0\|_F \leq C_2$, $\|\mathbf{Y}_s - \mathbf{Y}_0\|_F \leq C_2$, then we have*

$$\left\|\text{vec}(\mathbf{P}_s\mathbf{Q}_s^\top)\right\| \leq C_3\theta^{2s}\|\mathbf{r}_0\|^2 \quad and \quad \|\eta(\mathbf{H}_0 - \mathbf{H}_s)\mathbf{r}_s\| \leq C_4\theta^s\|\mathbf{r}_0\|$$

*for some constants $C_3$ and $C_4$ depending on $C_1$ and $C_2$. Moreover, if $C_1$ and $C_2$ satisfy*

$$(\max(\|\mathbf{X}_0\|, \|\mathbf{Y}_0\|) + C_2)\eta C_1\|\mathbf{r}_0\| \leq (1 - \theta)C_2, \tag{6}$$

*then we have $\|\mathbf{X}_{t+1} - \mathbf{X}_0\|_F \leq C_2$ and $\|\mathbf{Y}_{t+1} - \mathbf{Y}_0\|_F \leq C_2$.*

Lemma 3 shows that $\|\boldsymbol{\xi}_t\| = O(\theta^t)$ if the residual shrinks exponentially and the iterates are not too far from initialization, which in turn implies that $\mathbf{X}_{t+1}$ and $\mathbf{Y}_{t+1}$ are also within the $C_2$-balls around their initialization. It turns out that there is a set of valid coefficients for the induction to go through as long as the $c$ in (2) is sufficiently large. Therefore, by choosing $c$ properly and plugging in $\rho = \frac{L-\mu}{L+\mu}$ and $\theta = 1 - \frac{\mu}{L}$, we prove the convergence rate for GD, and the iteration complexity follows immediately from Proposition 1. The complete proof is provided in Appendix B.6.

**Remark 1.** *In our proof, the unbalanced initialization guarantees the existence of induction constants in Lemma 3. The amount of unbalance affects the constant factors but will not affect the convergence rate $(1 - \frac{\mu}{L})^t$. To be explicit, suppose we initialize $\mathbf{X}_0 = c_1 \mathbf{A} \mathbf{\Phi}_1 \in \mathbb{R}^{m \times d}$, $\mathbf{Y}_0 = c_2 \mathbf{\Phi}_2 \in \mathbb{R}^{n \times d}$, where $[\mathbf{\Phi}_1]_{i,j} \sim N(0, 1/d)$ and $[\mathbf{\Phi}_2]_{i,j} \sim N(0, 1/n)$, then by replacing $\mathbf{H}_0$ in Proposition 2 with $\mathbf{H}_0' = \mathbf{I}_n \otimes (\mathbf{X}_0 \mathbf{X}_0^\top)$, we can generalize the proof and obtain the same convergence rate when $c_1$ is sufficiently large and $c_1 c_2 = O(1)$. Meanwhile, we have $\|\mathbf{R}_0\|_F \leq (1 + O(c_1 c_2)) \|\mathbf{A}\|_F$ with high probability [Ward and Kolda, 2023]. Therefore, when $c_1$ is fixed, a smaller $c_2$ yields a smaller initial loss, resulting in a smaller constant factor. Meanwhile, the convergence rate remains the same as the condition number of $\mathbf{H}_0'$ is not affected, and the shift $\mathbf{H}_t - \mathbf{H}_0'$ is controlled for small $c_2$.*

## 3.2 Proof Sketch for NAG Convergence Rate (Theorem 2)

We now turn to prove Theorem 2. Similar to GD, we track the residual dynamics of NAG.

**Proposition 3** (NAG dynamics). *Let $\mathbf{P}_t = \mathbf{X}_{t+1} - \mathbf{X}_t$ and $\mathbf{Q}_t = \mathbf{Y}_{t+1} - \mathbf{Y}_t$ denote the update steps for $t \geq 0$. Then NAG (4) admits the following dynamics:*

$$
\begin{pmatrix} \mathbf{r}_{t+1} \\ \mathbf{r}_t \end{pmatrix} = \begin{pmatrix} (1+\beta)(\mathbf{I}_{mn} - \eta \mathbf{H}_0) & -\beta(\mathbf{I}_{mn} - \eta \mathbf{H}_0) \\ \mathbf{I}_{mn} & 0 \end{pmatrix} \begin{pmatrix} \mathbf{r}_t \\ \mathbf{r}_{t-1} \end{pmatrix} + \begin{pmatrix} \boldsymbol{\xi}_t \\ 0 \end{pmatrix}, \tag{7}
$$

*where $\mathbf{H}_t = (\mathbf{Y}_t \mathbf{Y}_t^\top) \otimes \mathbf{I}_m + \mathbf{I}_n \otimes (\mathbf{X}_t \mathbf{X}_t^\top)$, $\boldsymbol{\xi}_t = \boldsymbol{\zeta}_t + \boldsymbol{\iota}_t$,*

$$
\boldsymbol{\zeta}_t = \text{vec}(\mathbf{P}_t \mathbf{Q}_t^\top) + \beta \, \text{vec}(\mathbf{P}_{t-1} \mathbf{Q}_{t-1}^\top) + \beta \eta \, \text{vec}(\mathbf{R}_{t-1} \mathbf{Y}_{t-1} \mathbf{Q}_{t-1}^\top + \mathbf{P}_{t-1} \mathbf{X}_{t-1}^\top \mathbf{R}_{t-1}),
$$

$$
\boldsymbol{\iota}_t = (1+\beta)\eta(\mathbf{H}_0 - \mathbf{H}_t)\mathbf{r}_t - \beta \eta (\mathbf{H}_0 - \mathbf{H}_{t-1})\mathbf{r}_{t-1}.
$$

As Proposition 3 shows, NAG dynamics (7) has additional momentum terms involving $\mathbf{P}_t$ and $\mathbf{Q}_t$. When $\beta = 0$, it reduces to the GD dynamics (5). The introduction of momentum terms allows the linear part in (7) to contract $\mathbf{r}_t$ and $\boldsymbol{\xi}_t$ faster. To be more explicit, let

$$
\mathbf{T}_{\text{NAG}} := \begin{pmatrix} (1+\beta)(\mathbf{I}_{mn} - \eta \mathbf{H}_0) & -\beta(\mathbf{I}_{mn} - \eta \mathbf{H}_0) \\ \mathbf{I}_{mn} & 0 \end{pmatrix} \tag{8}
$$

denote the linear part of the system. The next lemma shows NAG improves the rate of contraction.

**Lemma 4** (NAG contractivity). *Let $\eta = \frac{1}{L}$, $\beta = \frac{\sqrt{L} - \sqrt{\mu}}{\sqrt{L} + \sqrt{\mu}}$, then for all $(\mathbf{u}, \mathbf{v}) \in \mathcal{H} \times \mathcal{H}$,*

$$
\left\| \mathbf{T}_{\text{NAG}} \begin{pmatrix} \mathbf{u} \\ \mathbf{v} \end{pmatrix} \right\| \leq \left( 1 - \sqrt{\frac{\mu}{L}} \right) \left\| \begin{pmatrix} \mathbf{u} \\ \mathbf{v} \end{pmatrix} \right\|.
$$

The price to pay for the faster rate of contraction is the additional perturbations. The $\boldsymbol{\iota}_t$ term characterizes dynamics shift, which can be controlled as GD in Lemma 3. The $\boldsymbol{\zeta}_t$ term characterizes higher-order terms in the dynamics (7), which can be controlled by the updates $\mathbf{P}_t$ and $\mathbf{Q}_t$. In GD, these terms correspond to the gradient so that they can be bounded if $\mathbf{R}_t$ shrinks and $\mathbf{X}_t$ and $\mathbf{Y}_t$ are not too far away from $\mathbf{X}_0$ and $\mathbf{Y}_0$. In NAG, we have

$$
\mathbf{P}_t = \eta \mathbf{R}_t \mathbf{Y}_t + \eta \sum_{s=1}^{t} \beta^{t-s+1} \mathbf{R}_s \mathbf{Y}_s,
$$

and a similar equation holds for $\mathbf{Q}_t$. If $\mathbf{R}_t$ shrinks at rate $\theta > \theta^2 \geq \beta$, then we have an $O(\theta^t)$ upper bound for $\|\mathbf{P}_t\|_F$ and $\|\mathbf{Q}_t\|_F$. We formalize the argument in the following induction lemma.

**Lemma 5.** *Suppose $0 < \beta \leq \theta^2 < \theta < 1$. If there exist some constants $C_1$ and $C_2$ such that for any $s \leq t$, the NAG dynamics (7) yields $\left\| \begin{pmatrix} \mathbf{r}_s \\ \mathbf{r}_{s-1} \end{pmatrix} \right\| \leq C_1 \theta^s \left\| \begin{pmatrix} \mathbf{r}_0 \\ \mathbf{r}_{-1} \end{pmatrix} \right\|$, $\|\mathbf{X}_s - \mathbf{X}_0\|_F \leq C_2$, and $\|\mathbf{Y}_s - \mathbf{Y}_0\|_F \leq C_2$, then we have*

$$
\|\boldsymbol{\zeta}_t\| \leq C_3 \theta^{2t} \left\| \begin{pmatrix} \mathbf{r}_0 \\ \mathbf{r}_{-1} \end{pmatrix} \right\|^2, \quad \text{and} \quad \|\boldsymbol{\iota}_t\| \leq C_4 \theta^t \left\| \begin{pmatrix} \mathbf{r}_0 \\ \mathbf{r}_{-1} \end{pmatrix} \right\|
$$

*for some constants $C_3$ and $C_4$ depending on $C_1$ and $C_2$. Moreover, if $C_1$ and $C_2$ satisfy*

$$
(\max(\|\mathbf{X}_0\|, \|\mathbf{Y}_0\|) + C_2) \eta C_1 \left\| \begin{pmatrix} \mathbf{r}_0 \\ \mathbf{r}_{-1} \end{pmatrix} \right\| \leq (1 - \theta)^2 C_2, \tag{9}
$$

*then we have $\|\mathbf{X}_{t+1} - \mathbf{X}_0\|_F \leq C_2$ and $\|\mathbf{Y}_{t+1} - \mathbf{Y}_0\|_F \leq C_2$.*

Lemma 5 is similar to Lemma 3. Again by choosing a sufficiently large $c$ to initialize $\mathbf{X}_0$, we can find a set of feasible coefficients for the induction. In particular, we plug in $\rho = 1 - \frac{\sqrt{\mu}}{\sqrt{L}}$, $\theta = 1 - \frac{\sqrt{\mu}}{2\sqrt{L}}$ and $\beta = \frac{\sqrt{L}-\sqrt{\mu}}{\sqrt{L}+\sqrt{\mu}}$, then $\underline{c}$ defined in Theorem 2 ensures the success of induction, hence the accelerated convergence rate of NAG is proved. The complete proof is provided in Appendix C.4.

**Remark 2.** *Our analysis differs from that of Ward and Kolda [2023]. Their analysis is based on the Polyak-Łojasiewicz (PL) inequality [Łojasiewicz, 1963]: $f(\mathbf{X}_t, \mathbf{Y})$ is approximately $\mu$-PL and $L$-smooth in $\mathbf{Y}$, and the unbalanced initialization (large $\mathbf{X}_0$ small $\mathbf{Y}_0$) ensures that only $\mathbf{Y}$ matters to the convergence rate, as $\mathbf{X}$ is not changing by much. Since the objective function in (1) is quadratic in $\mathbf{X}$, the problem has condition number $\hat{\kappa} := \frac{L}{\mu} = O(\kappa^2)$. With these notations, the complexity in Ward and Kolda [2023] reads as $O(\hat{\kappa} \log \frac{1}{\epsilon})$, which is standard for PL functions.*

*However, PL inequality cannot fully capture the properties of (1), and the analysis in Ward and Kolda [2023] does not apply to the case where $\mathbf{X}_t$ and $\mathbf{Y}_t$ are updated simultaneously rather than alternatingly. In fact, if we fix $\mathbf{X} \equiv \mathbf{X}_0$ and optimize $\mathbf{Y}$ only, then our initialization (2) makes the problem quasi-strongly convex (QSC), which is strictly stronger than PL [Necoara et al., 2019]. For QSC functions, NAG can achieve $O(\sqrt{\hat{\kappa}} \log \frac{1}{\epsilon})$ convergence rate Necoara et al. [2019], while for PL functions the rate can only be $\Omega(\hat{\kappa} \log \frac{1}{\epsilon})$ [Yue et al., 2023].*

*We note that simultaneously optimizing $\mathbf{X}$ and $\mathbf{Y}$ causes the nonconvexity issue and hence (1) does not fit in the framework for QSC functions as it requires convexity. Our results in Theorems 1 and 2 match the ones for QSC functions and Theorem 2 further matches the lower bound for general smooth strongly convex functions [Nemirovski and Yudin, 1983], which generally exhibit more favorable properties than nonconvex optimization problems to which (1) belongs. Hence, we conjecture that our rate bounds are tight for both GD and NAG. However, rigorous theory is yet to be constructed to solidify our conjecture.*

## 4 Extension to Linear Neural Network

Our analysis can be extended to the squared loss training of two-layer linear neural networks, which is equivalent to the following optimization problem:

$$\min_{\mathbf{X} \in \mathbb{R}^{m \times d}, \mathbf{Y} \in \mathbb{R}^{n \times d}} f(\mathbf{X}, \mathbf{Y}) = \frac{1}{2} \left\| \mathbf{L} - \mathbf{X}\mathbf{Y}^\top \mathbf{D} \right\|_{\mathrm{F}}^2. \tag{10}$$

Here, $\mathbf{D} \in \mathbb{R}^{n \times N}$ corresponds to all input data concatenated together, $\mathbf{L} \in \mathbb{R}^{m \times N}$ denotes the labels, $N$ is the total number of training data samples, and $d$ is the network width. We make the following interpolation assumption, which is commonly adopted in the study of the convergence rate of linear neural networks [Du and Hu, 2019, Hu et al., 2020, Wang et al., 2021].

**Assumption 1** (Interpolation). *There is $\mathbf{A}$ with $\mathrm{cond}(\mathbf{A}) = O(1)$ such that $\mathbf{L} = \mathbf{AD}$, $\mathrm{rank}(\mathbf{L}) = r$.*

Under Assumption 1, we can establish a linear convergence rate for NAG when the initialization is sufficiently unbalanced and $\mathbf{X}_0$ contains the column space of $\mathbf{L}$.

**Theorem 3.** *Let $\tilde{L} = \sigma_1^2(\mathbf{X}_0) \cdot \lambda_{\max}(\mathbf{DD}^\top)$, $\tilde{\mu} = \sigma_r^2(\mathbf{X}_0) \cdot \lambda_{\min}(\mathbf{DD}^\top)$. Suppose $\mathbf{Y}_0 = 0$, $\mathbf{X}_0$ is initialized such that $\mathrm{col}(\mathbf{X}_0) \supseteq \mathrm{col}(\mathbf{L})$ and it satisfies*

$$\tilde{\mu} p \geq 4\sqrt{2} \left\| \mathbf{LD}^\top \right\|_{\mathrm{F}} (1 + p), \tag{11}$$

*where $p = \frac{\sqrt{\tilde{\mu}}}{144\sqrt{\tilde{L}}}$ does not depend on the scaling of $\mathbf{X}_0$. If we choose $\eta = \frac{1}{\tilde{L}}$ and $\beta = \frac{\sqrt{\tilde{L}}-\sqrt{\tilde{\mu}}}{\sqrt{\tilde{L}}+\sqrt{\tilde{\mu}}}$, then the $t$-th iterate of NAG ($\mathbf{X}_t$ and $\mathbf{Y}_t$) will correspond to residual $\mathbf{R}_t = \mathbf{X}_t \mathbf{Y}_t^\top \mathbf{D} - \mathbf{L}$ satisfying*

$$\|\mathbf{R}_t\|_{\mathrm{F}} \leq \frac{\sigma_r^2(\mathbf{X}_0)\sigma_{\min}(\mathbf{D})}{576} \left( 1 - \frac{\sqrt{\tilde{\mu}}}{2\sqrt{\tilde{L}}} \right)^t.$$

*Equivalently, let $C = \frac{\sigma_r^2(\mathbf{X}_0)\sigma_{\min}(\mathbf{D})}{576\|\mathbf{LD}^\top\|_{\mathrm{F}}}$, then the iteration complexity for $\epsilon$ relative error is*

$$T = O\left( \frac{\sigma_1(\mathbf{X}_0)\sqrt{\lambda_{\max}(\mathbf{DD}^\top)}}{\sigma_r(\mathbf{X}_0)\sqrt{\lambda_{\min}(\mathbf{DD}^\top)}} \log \left( \frac{C}{\epsilon} \right) \right).$$

As Theorem 3 shows, if our initialization guarantees the column space of $\mathbf{X}_0$ contains columns of $\mathbf{L}$, then the residual shrinks at a linear rate. In the worst case, the columns of $\mathbf{L}$ span the whole space of $\mathbb{R}^m$, hence $d$ should be at least $m$. However, when the data exhibits some low-dimensional properties, e.g., $\mathbf{D}$ is low-rank, then $r$ can be much smaller than $m$ and $N$. In this case, an initialization similar to (2) can meet the requirement of Theorem 3. Moreover, note that the convergence rate depends on both $\mathbf{D}$ and $\mathbf{X}_0$, hence by orthonormalization we can make $\mathrm{cond}(\mathbf{X}_0) = 1$ for a faster rate. When $r \leq d \ll \min(m, N)$, such orthonormalization is affordable as it takes $O(md^2)$ time rather than $O(mN^2)$ in the worst case. We summarize these initialization options:

$$d \geq r, \quad \boldsymbol{\Phi} \in \mathbb{R}^{N \times d}, \quad [\boldsymbol{\Phi}]_{i,j} \sim \mathcal{N}(0, 1/d), \quad \mathbf{X}_0 = c \cdot \mathbf{L}\boldsymbol{\Phi}, \quad \mathbf{Y}_0 = 0; \tag{12}$$

$$d \geq r, \quad \boldsymbol{\Phi} \in \mathbb{R}^{N \times d}, \quad [\boldsymbol{\Phi}]_{i,j} \sim \mathcal{N}(0, 1/d), \quad \mathbf{X}_0 = c \cdot \mathsf{Orth}(\mathbf{L}\boldsymbol{\Phi}), \quad \mathbf{Y}_0 = 0; \tag{13}$$

$$d \geq m, \quad \boldsymbol{\Phi} \in \mathbb{R}^{m \times d}, \quad [\boldsymbol{\Phi}]_{i,j} \sim \mathcal{N}(0, 1/d), \quad \mathbf{X}_0 = c \cdot \boldsymbol{\Phi}, \quad \mathbf{Y}_0 = 0; \tag{14}$$

Here, $\mathsf{Orth}(\cdot)$ denotes the orthonormalization result whose columns are orthonormal. By applying singular value bounds and invoking Theorem 3, we obtain the following corollaries.

**Corollary 1.** *Suppose initialization* (12) *is applied with some sufficiently large $c$. For any $0 < \tau < c_1$, $0 < \delta < 1$, if $d \geq r - 1 + \Omega(\log \frac{1}{\delta})$, then with probability at least $1 - \delta$, NAG finds $\mathbf{X}_T$ and $\mathbf{Y}_T$ such that $f(\mathbf{X}_T, \mathbf{Y}_T) \leq \epsilon \|\mathbf{L}\mathbf{D}^\top\|_{\mathrm{F}}^2$ where*

$$T = O\left( \frac{d \cdot \mathrm{cond}(\mathbf{L})}{\tau(d - r + 1)} \frac{\sqrt{\lambda_{\max}(\mathbf{D}\mathbf{D}^\top)}}{\sqrt{\lambda_{\min}(\mathbf{D}\mathbf{D}^\top)}} \log \frac{1}{\epsilon} \right).$$

**Corollary 2.** *Suppose initialization* (13) *is applied with some sufficiently large $c$. If $d \geq r$, then with probability 1, NAG finds $\mathbf{X}_T$ and $\mathbf{Y}_T$ such that $f(\mathbf{X}_T, \mathbf{Y}_T) \leq \epsilon \|\mathbf{L}\mathbf{D}^\top\|_{\mathrm{F}}^2$ where*

$$T = O\left( \sqrt{\frac{\lambda_{\max}(\mathbf{D}\mathbf{D}^\top)}{\lambda_{\min}(\mathbf{D}\mathbf{D}^\top)}} \log \frac{1}{\epsilon} \right).$$

**Corollary 3.** *Suppose initialization* (14) *is applied with some sufficiently large $c$. For any $0 < \tau < c_1$, $0 < \delta < 1$, if $d \geq m - 1 + \Omega(\log \frac{1}{\delta})$, then with probability at least $1 - \delta$, NAG finds $\mathbf{X}_T$ and $\mathbf{Y}_T$ such that $f(\mathbf{X}_T, \mathbf{Y}_T) \leq \epsilon \|\mathbf{L}\mathbf{D}^\top\|_{\mathrm{F}}^2$ where*

$$T = O\left( \frac{d}{\tau(d - m + 1)} \frac{\sqrt{\lambda_{\max}(\mathbf{D}\mathbf{D}^\top)}}{\sqrt{\lambda_{\min}(\mathbf{D}\mathbf{D}^\top)}} \log \frac{1}{\epsilon} \right).$$

**Remark 3.** *While we only consider NAG in this section, our analysis can be directly applied to GD and obtain $O\left( \frac{\sigma_1^2(\mathbf{X}_0)\lambda_{\max}(\mathbf{D}\mathbf{D}^\top)}{\sigma_r^2(\mathbf{X}_0)\lambda_{\min}(\mathbf{D}\mathbf{D}^\top)} \log \frac{1}{\epsilon} \right)$ convergence rate with initializations* (12) *to* (14).

Corollaries 2 and 3 show accelerated convergence rate of NAG, as their dependence on the condition number $\kappa := \frac{\lambda_{\max}(\mathbf{D}\mathbf{D}^\top)}{\lambda_{\min}(\mathbf{D}\mathbf{D}^\top)} = \mathrm{cond}^2(\mathbf{D})$ is $O(\sqrt{\kappa})$ rather than $O(\kappa)$, matching the results in Wang et al. [2021] for HB and Liu et al. [2022] for NAG. Meanwhile, Corollary 1 has an additional dependence on $\mathrm{cond}(\mathbf{L})$. Under Assumption 1, $\mathrm{cond}(\mathbf{L}) = O(\sqrt{\kappa})$ and hence the overall dependence is $O(\kappa)$. Although this is slower than NAG with initialization (13) or (14), it still outperforms GD with initialization (12), which has $O(\kappa^2)$ dependence. Compared to previous results listed in Table 1, we only require the network width to be $\Omega(r + \log \frac{1}{\delta})$ or $\Omega(m + \log \frac{1}{\delta})$ depending on the initialization and there is no additional dependence on the input rank or condition number. When the data is low-rank, NAG with initialization (12) enables the sublinear-width (w.r.t. output dimension and sample size) network to converge linearly. It can be further accelerated if orthonormalization is adopted (13), which echos the orthogonal initialization in Hu et al. [2020], Wang et al. [2021]. In the general case, our analysis still provides a tighter result, as (14) only requires the width to be $\Omega(m + \log \frac{1}{\delta})$.

## 5  Numerical Experiment

We validate our results via numerical experiments. For matrix factorization (1), we construct $\mathbf{A} = \mathbf{U}\boldsymbol{\Sigma}\mathbf{V}^\top \in \mathbb{R}^{100 \times 80}$, where $\boldsymbol{\Sigma} \in \mathbb{R}^{5 \times 5}$ is diagonal with $\sigma_1(\boldsymbol{\Sigma}) = 1$ and $\sigma_5(\boldsymbol{\Sigma}) = 0.2$, and

**U** and **V** are orthonormal matrices. We set different levels of overparameterization ($d \geq 5$) and initialize $\mathbf{X}_0$ and $\mathbf{Y}_0$ according to (2) with $c = 50\sqrt{d}$. For linear neural network (10), we construct the input data matrix $\mathbf{D} = \mathbf{U}\mathbf{\Sigma}\mathbf{V}^\top \in \mathbb{R}^{80 \times 120}$, where $\mathbf{\Sigma} \in \mathbb{R}^{5 \times 5}$ is diagonal with $\sigma_1(\mathbf{\Sigma}) = 1$ and $\sigma_5(\mathbf{\Sigma}) = 0.5$, **U** is orthonormal and **V** is Gaussian. We use a Gaussian matrix $\mathbf{A} \in \mathbb{R}^{100 \times 80}$ to construct the label matrix $\mathbf{L} = \mathbf{A}\mathbf{D}$. We keep $c = 50\sqrt{d}$ and initialize $\mathbf{X}_0$ and $\mathbf{Y}_0$ according to (12). We run all experiments with 10 different initialization seeds and take the average.

We first compare GD and AltGD. For matrix factorization, We use the same initialization and the same step size $\eta = 2/(L + \mu)$, where $L$ and $\mu$ are computed as defined in Theorems 1 and 2. For linear neural networks, $L$ and $\mu$ are replaced by $\tilde{L}$ and $\tilde{\mu}$ in Theorem 3. As shown in Figure 1, they perform very similarly and the loss curves are overlapped. To better illustrate, we additionally use $\eta = 1/L$ for GD, and it performs differently from GD/AltGD with $\eta = 2/(L + \mu)$.

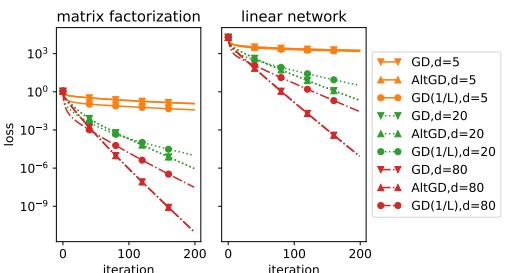

Figure 1: *GD and AltGD achieve similar performance. The left plot is for* (1)*, and the right plot is for* (10)*.*

We then compare GD and NAG. For matrix factorization, we use $\eta = 2/(L+\mu)$ for GD and use $\eta = 1/L$ and $\beta = (\sqrt{L} - \sqrt{\mu})/(\sqrt{L} + \sqrt{\mu})$ for NAG, where $L$ and $\mu$ are computed as defined in Theorem 2. For linear neural networks, we replace $L$ and $\mu$ by $\tilde{L}$ and $\tilde{\mu}$ defined in Theorem 3. The results are shown in Figure 2. As illustrated, NAG exhibits much faster convergence than GD. Moreover, a higher overparameterization level helps accelerate convergence, as predicted by the prefactor $O(\text{poly}(d(d - r + 1)^{-1}))$ in our iteration complexity.

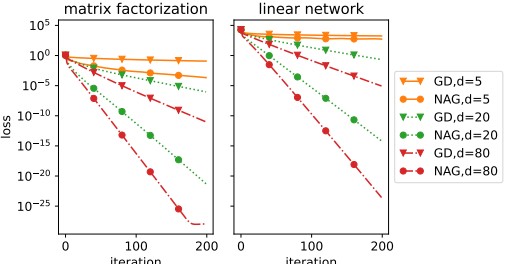

Figure 2: *NAG converges faster than GD. The left plot is for* (1)*, and the right plot is for* (10)*.*

To further illustrate the tightness of our theory, we compare our theoretical predictions with the actual loss in matrix factorization, as shown in Figure 3. We set $c = 200\sqrt{d}$ and $\sigma_5(\mathbf{\Sigma}) \in \{0.1, 0.01\}$, keeping other settings unchanged. The theoretical prediction at step $t$ is computed as $(1 - \mu/L)^{2t} \cdot f(\mathbf{X}_0, \mathbf{Y}_0)$ for GD and $(1 - \sqrt{\mu}/(2\sqrt{L}))^{2t} \cdot f(\mathbf{X}_0, \mathbf{Y}_0)$ for NAG. We observe that the slope of the predicted loss closely matches the actual loss, supporting the tightness of our theory, especially for GD. Additional experiments are provided in Appendix E.

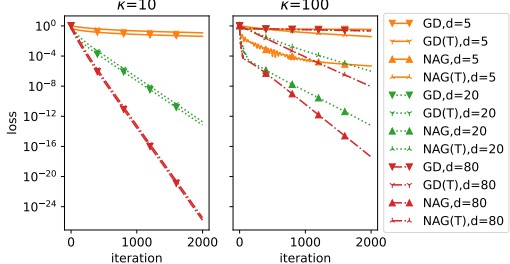

Figure 3: *Comparison of predicted loss and numerical loss for matrix factorization. The left plot is for GD where $\kappa = 10$, and the right plot is for GD and NAG where $\kappa = 100$. (T) denotes theory prediction.*

## 6 Conclusion and Future Work

We establish the convergence rate of GD and NAG for rectangular matrix factorization (1) under an unbalanced initialization and show the provable acceleration of NAG. We further extend our analysis to linear neural networks (10) and show the acceleration of NAG without excessive width requirements in previous work. Numerical experiments are provided to support our theory.

We believe our analysis can be extended to initialization where $\mathbf{X}_0 \approx c\mathbf{A}\mathbf{\Phi}$ and $\mathbf{Y}_0 \approx 0$ rather than exact equalities. Relaxing the exact rank-$r$ condition to approximately rank-$r$ is also a possible generalization. The linear neural network model considered in this paper cannot fully capture the practical settings. We leave the extension to nonlinear activations for future work.

## Acknowledgments and Disclosure of Funding

The authors are grateful for the partial support by NSF DMS-1847802, Cullen-Peck Scholarship, and GT-Emory Humanity.AI Award. RW was supported in part by NSF DMS-1952735, NSF IFML grantv2019844, NSF DMS-N2109155, and NSF 2217033. We thank the anonymous reviewers for their helpful comments.

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

# A Singular Value Bounds

## A.1 Singular Value Bounds for Random Matrix

**Proposition 4** (Rudelson and Vershynin [2009]). *Let $\mathbf{A}$ be an $N \times n$ random matrix, $N \geq n$, whose elements are i.i.d. zero mean sub-Gaussian random variables with unit variance. Then for $\tau \geq 0$, we have*

$$\mathbb{P}\big(\sigma_n(\mathbf{A}) \leq \tau(\sqrt{N} - \sqrt{n-1})\big) \leq (c_1\tau)^{N-n+1} + e^{-c_2 N}$$

*where $c_1, c_2 > 0$ depend (polynomially) only on the sub-Gaussian moment.*

**Proposition 5** (Vershynin [2010]). *Let $\mathbf{A}$ be an $N \times n$ random matrix, $N \geq n$, whose elements are i.i.d. zero mean Gaussian random variables with unit variance. Then for $t \geq 0$, we have*

$$\mathbb{P}\big(\sigma_1(\mathbf{A}) \geq \sqrt{N} + \sqrt{n} + t\big) \leq e^{-\frac{t^2}{2}}.$$

## A.2 Proof of Proposition 1

*Proof of Proposition 1.* Singular value decompose $\mathbf{A}$ as $\mathbf{A} = \mathbf{U}\boldsymbol{\Sigma}\mathbf{V}^\top$, then $\mathbf{X}_0 = c\mathbf{U}\boldsymbol{\Sigma}\mathbf{V}^\top\boldsymbol{\Phi}$. Since $\mathbf{V}^\top\mathbf{V} = \mathbf{I}_r$, the columns of $\mathbf{V}^\top\boldsymbol{\Phi} \in \mathbb{R}^{r \times d}$ are independent Gaussian vectors with distribution $\mathcal{N}(0, \frac{1}{d}\mathbf{V}^\top\mathbf{V}) = \mathcal{N}(0, \frac{1}{d}\mathbf{I}_r)$. By Proposition 4 in Appendix A, we have

$$\mathbb{P}\left(\sigma_r(\mathbf{V}^\top\boldsymbol{\Phi}) \leq \tau\left(1 - \frac{\sqrt{r-1}}{\sqrt{d}}\right)\right) \leq e^{-(d-r+1)\log\frac{1}{c_1\tau}} + e^{-c_2 d}$$

for some universal constants $c_1$ and $c_2$ and any $\tau \geq 0$. On the other hand, by Proposition 5 in Appendix A, we have

$$\mathbb{P}\left(\sigma_1(\mathbf{V}^\top\boldsymbol{\Phi}) \geq \frac{\sqrt{d} + \sqrt{r} + \sqrt{s}}{\sqrt{d}}\right) \leq e^{-\frac{s}{2}}.$$

Plugging in $s = d$ and applying the union bound yield

$$\mathbb{P}\left(\frac{\tau(\sqrt{d} - \sqrt{r-1})}{\sqrt{d}} \leq \sigma_r(\mathbf{V}^\top\boldsymbol{\Phi}) \leq \sigma_1(\mathbf{V}^\top\boldsymbol{\Phi}) \leq \frac{2\sqrt{d} + \sqrt{r}}{\sqrt{d}}\right) \geq 1 - \delta,$$

where $\delta = 3e^{-\min\{(d-r+1)\log\frac{1}{c_1\tau}, c_2 d, \frac{d}{2}\}}$. The proposition follows immediately from the fact that

$$c \cdot \sigma_r(\mathbf{V}^\top\boldsymbol{\Phi})\sigma_r(\mathbf{A}) \leq \sigma_r(\mathbf{X}_0) \leq \sigma_1(\mathbf{X}_0) \leq c \cdot \sigma_1(\mathbf{V}^\top\boldsymbol{\Phi})\sigma_1(\mathbf{A}).$$

$\square$

# B Missing Proofs for GD

## B.1 Auxiliary Lemma

**Lemma 6.** *Suppose $\{a_t\}_{t \geq 0}$ and $\{b_t\}_{t \geq 0}$ are two non-negative sequences satisfying*

$$a_{t+1} \leq \rho \cdot a_t + b_t, \quad b_t \leq \theta^t \cdot c_0,$$

*where $0 \leq \rho < \theta < 1$, $c_0 \geq 0$, then the following holds for all $t \geq 0$:*

$$a_t \leq \theta^t \cdot \left(a_0 + \frac{c_0}{\theta - \rho}\right).$$

*Proof.* The inequality holds trivially for $t = 0$. For $t \geq 0$, we have

$$\begin{aligned}
a_{t+1} &= \rho^{t+1} \cdot a_0 + \sum_{s=0}^{t} \rho^{t-s}\theta^s \cdot c_0 \\
&= \rho^{t+1} \cdot a_0 + \frac{\theta^{t+1} - \rho^{t+1}}{\theta - \rho} \cdot c_0 \\
&= \theta^{t+1} \cdot \left(a_0 + \frac{1}{\theta - \rho} \cdot c_0\right).
\end{aligned}$$

$\square$

## B.2 Proof of Proposition 2

*Proof of Proposition 2.* According to (3), we have

$$\mathbf{R}_{t+1} = \mathbf{X}_{t+1}\mathbf{Y}_{t+1}^\top - \mathbf{A}$$
$$= (\mathbf{X}_t + \mathbf{P}_t)(\mathbf{Y}_t + \mathbf{Q}_t)^\top - \mathbf{A}$$
$$= \mathbf{R}_t - \eta\left(\mathbf{R}_t\mathbf{Y}_t\mathbf{Y}_t^\top + \mathbf{X}_t\mathbf{X}_t^\top\mathbf{R}_t\right) + \mathbf{P}_t\mathbf{Q}_t^\top.$$

Applying vectorization on both sides yields

$$\mathbf{r}_{t+1} = \mathbf{r}_t - \eta\mathbf{H}_t\mathbf{r}_t + \beta(\mathbf{r}_t - \mathbf{r}_{t-1}) + \mathrm{vec}(\mathbf{P}_t\mathbf{Q}_t^\top)$$
$$= (\mathbf{I}_{mn} - \eta\mathbf{H}_t)\mathbf{r}_t + \mathrm{vec}(\mathbf{P}_t\mathbf{Q}_t^\top).$$

Hence we have the result. $\qquad\square$

## B.3 Proof of Lemma 1

*Proof of Lemma 1.* By Proposition 1, the symmetric matrix $\mathbf{H}_0 = \mathbf{I}_n \otimes (\mathbf{X}_0\mathbf{X}_0^\top)$ has $nr$ positive eigenvalues, and the eigensubspace of these positive eigenvalues is

$$\mathcal{H} = \prod_{i=1}^{n} \mathrm{col}(\mathbf{X}_0) = \prod_{i=1}^{n} \mathrm{col}(\mathbf{A}).$$

According to the GD update (3),

$$\mathrm{col}(\mathbf{X}_{t+1}) \subseteq \mathrm{col}(\mathbf{X}_t) + \mathrm{col}(\mathbf{X}_t\mathbf{Y}_t^\top\mathbf{Y}_t) + \mathrm{col}(\mathbf{A}\mathbf{Y}_t) \subseteq \mathrm{col}(\mathbf{X}_t) + \mathrm{col}(\mathbf{A}),$$

hence by induction we conclude $\mathrm{col}(\mathbf{X}_t) \subseteq \mathrm{col}(\mathbf{A})$ for all $t \geq 0$. As a result, we have

$$\mathbf{r}_t = \mathrm{vec}(\mathbf{X}_t\mathbf{Y}_t^\top - \mathbf{A}) \in \mathcal{H}.$$

For $\boldsymbol{\xi}_t$, notice that

$$\mathrm{col}(\mathbf{R}_t\mathbf{Y}_t\mathbf{Y}_t^\top + \mathbf{X}_t\mathbf{X}_t^\top\mathbf{R}_t) \subseteq \mathrm{col}(\mathbf{R}_t) + \mathrm{col}(\mathbf{X}_t) \subseteq \mathrm{col}(\mathbf{A})$$

and

$$\mathrm{col}(\mathbf{P}_t\mathbf{Q}_t^\top) = \mathrm{col}((\mathbf{X}_{t+1} - \mathbf{X}_t)(\mathbf{Y}_{t+1} - \mathbf{Y}_t)^\top) \subseteq \mathrm{col}(\mathbf{X}_{t+1}) + \mathrm{col}(\mathbf{X}_t) \subseteq \mathrm{col}(\mathbf{A}),$$

thus we have

$$\boldsymbol{\xi}_t = \eta \cdot \mathrm{vec}(\mathbf{R}_t\mathbf{Y}_0\mathbf{Y}_0^\top + \mathbf{X}_0\mathbf{X}_0^\top\mathbf{R}_t - \mathbf{R}_t\mathbf{Y}_t\mathbf{Y}_t^\top - \mathbf{X}_t\mathbf{X}_t^\top\mathbf{R}_t) + \mathrm{vec}(\mathbf{P}_t\mathbf{Q}_t^\top) \in \mathcal{H}.$$

$\qquad\square$

## B.4 Proof of Lemma 2

*Proof of Lemma 2.* Since $\mathbf{I}_{mn}$ commutes with symmetric matrix $\mathbf{H}_0$, we can simultaneously diagonalize the two matrices and get

$$\lambda_i(\mathbf{T}_{\mathrm{GD}}) = 1 - \eta\lambda_{mn-i}(\mathbf{H}_0), \quad \forall i = 1, 2, \ldots, mn.$$

When $\eta \in (0, \frac{2}{L})$, $\lambda_i(\mathbf{T}_{\mathrm{GD}}) = 1$ for $i = 1, 2, \ldots, (m-r)n$. Let $\{\mathbf{v}_i\}_{i=1}^{mn}$ be orthonormal eigenvectors, $\mathbf{v}_i$ corresponds to $\lambda_i(\mathbf{T}_{\mathrm{GD}})$, then we have $\mathrm{Span}(\{\mathbf{v}_i\}_{i=1}^{(m-r)n}) = \ker(\mathbf{H}_0) \perp \mathcal{H}$. Consequently,

$$\|\mathbf{T}_{\mathrm{GD}}\mathbf{v}\| = \left\|\mathbf{T}_{\mathrm{GD}}\left(\sum_{i=1}^{mn}\langle\mathbf{v}, \mathbf{v}_i\rangle\mathbf{v}_i\right)\right\|$$
$$= \sqrt{\sum_{i=(m-r)n+1}^{mn}\langle\mathbf{v}, \mathbf{v}_i\rangle^2\lambda_i^2(\mathbf{T}_{\mathrm{GD}})}$$
$$\leq \max_{(m-r)n+1\leq i\leq mn}|\lambda_i(\mathbf{T}_{\mathrm{GD}})|\,\|\mathbf{v}\|$$
$$= \max\{|1 - \eta L|, |1 - \eta\mu|\}\,\|\mathbf{v}\|,$$

where the second identity is from $\mathbf{v} \in \mathcal{H} \perp \ker(\mathbf{H}_0)$. Plugging in the step size yields the second result. $\qquad\square$

## B.5 Proof of Lemma 3

*Proof of Lemma 3.* For all $s \le t$, by assumption we have

$$
\begin{aligned}
\|\mathbf{P}_s\|_{\mathrm{F}} &= \eta \|\mathbf{R}_s \mathbf{Y}_s\|_{\mathrm{F}} \\
&\le \eta \|\mathbf{Y}_s\| \|\mathbf{R}_s\|_{\mathrm{F}} \\
&\le \eta(\|\mathbf{Y}_0\| + \|\mathbf{Y}_s - \mathbf{Y}_0\|) \|\mathbf{R}_s\|_{\mathrm{F}} \\
&\le \eta(\|\mathbf{Y}_0\| + \|\mathbf{Y}_s - \mathbf{Y}_0\|_{\mathrm{F}}) \|\mathbf{R}_s\|_{\mathrm{F}} \\
&\le \eta(\|\mathbf{Y}_0\| + C_2) \|\mathbf{R}_s\|_{\mathrm{F}} \\
&\le \eta(\|\mathbf{Y}_0\| + C_2) C_1 \theta^s \|\mathbf{r}_0\| .
\end{aligned}
$$

Similarly, we have

$$
\|\mathbf{Q}_s\|_{\mathrm{F}} \le \eta(\|\mathbf{X}_0\| + C_2) C_1 \theta^s \|\mathbf{r}_0\| .
$$

Combining the two bounds yields

$$
\|\mathrm{vec}(\mathbf{P}_s \mathbf{Q}_s^\top)\| = \|\mathbf{P}_s \mathbf{Q}_s^\top\|_{\mathrm{F}} \le \|\mathbf{P}_s\|_{\mathrm{F}} \|\mathbf{Q}_s\|_{\mathrm{F}} \le C_3 \theta^{2t} \|\mathbf{r}_0\|^2 ,
$$

where $C_3 = \eta^2 C_1^2 (\|\mathbf{X}_0\| + C_2)(\|\mathbf{Y}_0\| + C_2)$.

For the second part, we have

$$
\begin{aligned}
\|(\mathbf{H}_0 - \mathbf{H}_s)\mathbf{r}_s\| &= \|\mathbf{R}_s(\mathbf{Y}_0 \mathbf{Y}_0^\top - \mathbf{Y}_s \mathbf{Y}_s^\top) + (\mathbf{X}_0 \mathbf{X}_0^\top - \mathbf{X}_s \mathbf{X}_s^\top)\mathbf{R}_s\|_{\mathrm{F}} \\
&\le \|\mathbf{R}_s(\mathbf{Y}_0 \mathbf{Y}_0^\top - \mathbf{Y}_s \mathbf{Y}_s^\top)\|_{\mathrm{F}} + \|(\mathbf{X}_0 \mathbf{X}_0^\top - \mathbf{X}_s \mathbf{X}_s^\top)\mathbf{R}_s\|_{\mathrm{F}} \\
&\le \|\mathbf{Y}_0 \mathbf{Y}_0^\top - \mathbf{Y}_s \mathbf{Y}_s^\top\| \|\mathbf{R}_s\|_{\mathrm{F}} + \|\mathbf{X}_0 \mathbf{X}_0^\top - \mathbf{X}_s \mathbf{X}_s^\top\| \|\mathbf{R}_s\|_{\mathrm{F}} \\
&\le (2\|\mathbf{Y}_0\| + \|\mathbf{Y}_s - \mathbf{Y}_0\|_{\mathrm{F}}) \|\mathbf{Y}_s - \mathbf{Y}_0\|_{\mathrm{F}} \|\mathbf{R}_s\|_{\mathrm{F}} \\
&\quad + (2\|\mathbf{X}_0\| + \|\mathbf{X}_s - \mathbf{X}_0\|_{\mathrm{F}}) \|\mathbf{X}_s - \mathbf{X}_0\|_{\mathrm{F}} \|\mathbf{R}_s\|_{\mathrm{F}} \\
&\le 2(\|\mathbf{X}_0\| + \|\mathbf{Y}_0\| + C_2)C_2 \|\mathbf{R}_s\|_{\mathrm{F}} \\
&\le C_4 \theta^s \|\mathbf{r}_0\| ,
\end{aligned}
$$

where $C_4 = 2\eta(\|\mathbf{X}_0\| + \|\mathbf{Y}_0\| + C_2)C_1 C_2$.

Finally, when (6) holds, we have

$$
\|\mathbf{X}_{t+1} - \mathbf{X}_0\|_{\mathrm{F}} \le \sum_{s=0}^{t} \|\mathbf{P}_s\|_{\mathrm{F}} \le \frac{\eta(\|\mathbf{Y}_0\| + C_2)C_1}{1 - \theta} \|\mathbf{r}_0\| \le C_2.
$$

Similarly, we have $\|\mathbf{Y}_{t+1} - \mathbf{Y}_0\|_{\mathrm{F}} \le C_2$. □

## B.6 Proof of Theorem 1

*Proof of Theorem 1.* Let $C_1$ to $C_4$ be constants defined in Lemma 3. Define $\rho = \frac{L-\mu}{L+\mu}$, $\theta = 1 - \frac{\mu}{L}$, $a_t = C_1 \|\mathbf{r}_t\|$, and $b_t = C_1 \|\boldsymbol{\xi}_t\|$ for $t \ge 0$. By Proposition 2 and lemmas 1 and 2 we have

$$
a_{t+1} \le \rho \cdot a_t + b_t
$$

for all $t \ge 0$. It remains to show that $b_t \le \theta^t \cdot c_0$. By initialization (2), $a_0 = C_1 \|\mathbf{r}_0\| = C_1 \|\mathbf{A}\|_{\mathrm{F}}$, $b_0 = 0$. Let $C_1 = \frac{\mu(L+\mu)p}{2\|\mathbf{A}\|_{\mathrm{F}} L(1+p)}$ and $C_2 = p\sqrt{L}$ where $p = \frac{\mu(L-\mu)}{24L^2} \in (0,1)$. Plugging $\eta = \frac{2}{L+\mu}$, $\|\mathbf{X}_0\| = \sqrt{L}$ and $\|\mathbf{Y}_0\| = 0$ into $C_3$ and $C_4$ yields

$$
C_3 = \frac{\mu^2 p^3}{\|\mathbf{A}\|_{\mathrm{F}}^2 L(1+p)}, \quad C_4 = \frac{2\mu p^2}{\|\mathbf{A}\|_{\mathrm{F}}} .
$$

Let

$$
c_0 = C_1(C_3 \|\mathbf{r}_0\| + C_4) \|\mathbf{r}_0\| ,
$$

then we can show the following relations:

$$
a_0 + \frac{c_0}{\theta - \rho} \le C_1^2 \|\mathbf{A}\|_{\mathrm{F}}, \quad C_1 \ge 1. \tag{15}
$$

Indeed, by Proposition 1, with probability at least $1 - \delta$, our choice of $c$ guarantees

$$\mu \geq \frac{144 \operatorname{cond}^4(\mathbf{X}_0) \|\mathbf{A}\|_{\mathrm{F}}}{(\operatorname{cond}^2(\mathbf{X}_0) - 1)} = \frac{144 L^2 \|\mathbf{A}\|_{\mathrm{F}}}{\mu(L - \mu)}. \tag{16}$$

Our goal is to show

$$a_0 + \frac{c_0}{\theta - \rho} = C_1 \|\mathbf{A}\|_{\mathrm{F}} + C_1 (C_3 \|\mathbf{A}\|_{\mathrm{F}} + C_4) \|\mathbf{A}\|_{\mathrm{F}} \cdot \frac{L(L + \mu)}{\mu(L - \mu)} \leq C_1^2 \|\mathbf{A}\|_{\mathrm{F}},$$

which is equivalent to

$$\|\mathbf{A}\|_{\mathrm{F}} + \left( \frac{\mu p^3}{L(1 + p)} + 2p^2 \right) \cdot \frac{L(L + \mu)}{L - \mu} \leq \frac{\mu(L + \mu)p}{2L(1 + p)}.$$

The above inequality holds when:

$$\|\mathbf{A}\|_{\mathrm{F}} \leq \frac{\mu(L + \mu)p}{6L(1 + p)}, \tag{17}$$

$$\frac{p^2}{L - \mu} \leq \frac{1}{6L}, \tag{18}$$

$$\frac{2pL}{L - \mu} \leq \frac{\mu}{6L(1 + p)}. \tag{19}$$

Let $p = \frac{\mu(L - \mu)}{24L^2}$, then we have $p < 1$, $pL < \mu$ and

$$\frac{p^2}{L - \mu} \leq \frac{p}{L - \mu} = \frac{\mu}{24L^2} \leq \frac{1}{6L},$$

$$\frac{2pL}{L - \mu} \leq \frac{\mu}{12L} \leq \frac{\mu}{6L(1 + p)},$$

thus (18) and (19) hold. Finally, (17) holds in view of (16):

$$\frac{\mu(L + \mu)p}{6L(1 + p)} \geq \frac{\mu p}{6} = \frac{\mu^2(L - \mu)}{144L^2} \geq \|\mathbf{A}\|_{\mathrm{F}}.$$

Combining the results proves the (15).

Now we can proceed with the induction in Lemma 3. Firstly, $\|\mathbf{r}_0\| \leq C_1 \|\mathbf{r}_0\|$ as $C_1 \geq 1$ by (15), and $\|\mathbf{X}_0 - \mathbf{X}_0\|_{\mathrm{F}} = \|\mathbf{Y}_0 - \mathbf{Y}_0\|_{\mathrm{F}} = 0 \leq C_2$. Suppose the induction conditions in Lemma 2 holds for $s \leq t$, then we have

$$b_s = C_1 \|\boldsymbol{\xi}_s\| \leq C_1 (C_3 \theta^{2s} \|\mathbf{r}_0\|^2 + C_4 \theta^s \|\mathbf{r}_0\|) \leq c_0 \cdot \theta^s.$$

Consequently, by Lemma 6 and (15) we have

$$a_{t+1} \leq \theta^{t+1} \cdot \left( a_0 + \frac{c_0}{\theta - \rho} \right) \leq C_1^2 \cdot \theta^{t+1} \|\mathbf{A}\|_{\mathrm{F}},$$

thus $\|\mathbf{r}_{t+1}\| \leq C_1 \theta^{t+1} \|\mathbf{r}_0\|$. Moreover, by our construction of $C_1$ and $C_2$, (6) always holds, thus we also have $\|\mathbf{X}_{t+1} - \mathbf{X}_0\|_{\mathrm{F}} \leq C_2$ and $\|\mathbf{Y}_{t+1} - \mathbf{Y}_0\|_{\mathrm{F}} \leq C_2$. All conditions for the $t + 1$ step are satisfied, hence the proof is completed by induction. Plugging in $C_1$ and the choice of $c$ yields the results. $\qquad \square$

# C  Missing Proofs for NAG

## C.1  Proof of Proposition 3

*Proof of Proposition 3.*  According to the NAG update rule, we have

$$
\begin{aligned}
\mathbf{R}_{t+1} &= \mathbf{X}_{t+1}\mathbf{Y}_{t+1}^\top - \mathbf{A} \\
&= (\mathbf{X}_t + \mathbf{P}_t)(\mathbf{Y}_t + \mathbf{Q}_t)^\top - \mathbf{A} \\
&= \mathbf{R}_t + \mathbf{P}_t\mathbf{Y}_t^\top + \mathbf{X}_t\mathbf{Q}_t^\top + \mathbf{P}_t\mathbf{Q}_t^\top \\
&= \mathbf{R}_t + \left(\beta(\mathbf{X}_t - \mathbf{X}_{t-1}) - (1+\beta)\eta\mathbf{R}_t\mathbf{Y}_t + \beta\eta\mathbf{R}_{t-1}\mathbf{Y}_{t-1}\right)\mathbf{Y}_t^\top \\
&\quad + \mathbf{X}_t\left(\beta(\mathbf{Y}_t^\top - \mathbf{Y}_{t-1}^\top) - (1+\beta)\eta\mathbf{X}_t^\top\mathbf{R}_t + \beta\eta\mathbf{X}_{t-1}^\top\mathbf{R}_{t-1}\right) + \mathbf{P}_t\mathbf{Q}_t^\top \\
&= \mathbf{R}_t - (1+\beta)\eta\left(\mathbf{R}_t\mathbf{Y}_t\mathbf{Y}_t^\top + \mathbf{X}_t\mathbf{X}_t^\top\mathbf{R}_t\right) + \beta(\mathbf{X}_t\mathbf{Y}_t^\top - \mathbf{X}_{t-1}\mathbf{Y}_{t-1}^\top) \\
&\quad + \beta\eta\left(\mathbf{R}_{t-1}\mathbf{Y}_{t-1}\mathbf{Y}_{t-1}^\top + \mathbf{X}_{t-1}\mathbf{X}_{t-1}^\top\mathbf{R}_{t-1}\right) + \beta(\mathbf{X}_t\mathbf{Y}_t^\top + \mathbf{X}_{t-1}\mathbf{Y}_{t-1}^\top) - \beta\left(\mathbf{X}_{t-1}\mathbf{Y}_t^\top + \mathbf{X}_t\mathbf{Y}_{t-1}^\top\right) \\
&\quad + \beta\eta\left(\mathbf{R}_{t-1}\mathbf{Y}_{t-1}\mathbf{Y}_t^\top + \mathbf{X}_t\mathbf{X}_{t-1}^\top\mathbf{R}_{t-1} - \mathbf{R}_{t-1}\mathbf{Y}_{t-1}\mathbf{Y}_{t-1}^\top - \mathbf{X}_{t-1}\mathbf{X}_{t-1}^\top\mathbf{R}_{t-1}\right) + \mathbf{P}_t\mathbf{Q}_t^\top \\
&= \mathbf{R}_t - (1+\beta)\eta\left(\mathbf{R}_t\mathbf{Y}_t\mathbf{Y}_t^\top + \mathbf{X}_t\mathbf{X}_t^\top\mathbf{R}_t\right) + \beta(\mathbf{R}_t - \mathbf{R}_{t-1}) \\
&\quad + \beta\eta\left(\mathbf{R}_{t-1}\mathbf{Y}_{t-1}\mathbf{Y}_{t-1}^\top + \mathbf{X}_{t-1}\mathbf{X}_{t-1}^\top\mathbf{R}_{t-1}\right) + \beta(\mathbf{X}_t\mathbf{Y}_t^\top + \mathbf{X}_{t-1}\mathbf{Y}_{t-1}^\top - \mathbf{X}_{t-1}\mathbf{Y}_t^\top - \mathbf{X}_t\mathbf{Y}_{t-1}^\top) \\
&\quad + \beta\eta\left(\mathbf{R}_{t-1}\mathbf{Y}_{t-1}\mathbf{Y}_t^\top + \mathbf{X}_t\mathbf{X}_{t-1}^\top\mathbf{R}_{t-1} - \mathbf{R}_{t-1}\mathbf{Y}_{t-1}\mathbf{Y}_{t-1}^\top - \mathbf{X}_{t-1}\mathbf{X}_{t-1}^\top\mathbf{R}_{t-1}\right) + \mathbf{P}_t\mathbf{Q}_t^\top .
\end{aligned}
$$

Applying vectorization on both sides yields

$$
\begin{aligned}
\mathbf{r}_{t+1} &= \mathbf{r}_t - (1+\beta)\eta\mathbf{H}_t\mathbf{r}_t + \beta(\mathbf{r}_t - \mathbf{r}_{t-1}) + \beta\eta\mathbf{H}_{t-1}\mathbf{r}_{t-1} \\
&\quad + \beta\,\mathrm{vec}(\mathbf{X}_t\mathbf{Y}_t^\top + \mathbf{X}_{t-1}\mathbf{Y}_{t-1}^\top - \mathbf{X}_{t-1}\mathbf{Y}_t^\top - \mathbf{X}_t\mathbf{Y}_{t-1}^\top) \\
&\quad + \beta\eta\,\mathrm{vec}(\mathbf{R}_{t-1}\mathbf{Y}_{t-1}\mathbf{Y}_t^\top + \mathbf{X}_t\mathbf{X}_{t-1}^\top\mathbf{R}_{t-1} - \mathbf{R}_{t-1}\mathbf{Y}_{t-1}\mathbf{Y}_{t-1}^\top - \mathbf{X}_{t-1}\mathbf{X}_{t-1}^\top\mathbf{R}_{t-1}) + \mathrm{vec}(\mathbf{P}_t\mathbf{Q}_t^\top) \\
&= (1+\beta)(\mathbf{I}_{mn} - \eta\mathbf{H}_t)\mathbf{r}_t - \beta(\mathbf{I}_{mn} - \eta\mathbf{H}_{t-1})\mathbf{r}_{t-1} + \boldsymbol{\psi}_t + \boldsymbol{\phi}_t.
\end{aligned}
$$

Hence we have

$$
\begin{pmatrix}\mathbf{r}_{t+1}\\ \mathbf{r}_t\end{pmatrix} = \begin{pmatrix}(1+\beta)(\mathbf{I}_{mn} - \eta\mathbf{H}_0) & -\beta(\mathbf{I}_{mn} - \eta\mathbf{H}_0)\\ \mathbf{I}_{mn} & 0\end{pmatrix}\begin{pmatrix}\mathbf{r}_t\\ \mathbf{r}_{t-1}\end{pmatrix} + \begin{pmatrix}\boldsymbol{\xi}_t\\ 0\end{pmatrix}.
$$

$\square$

## C.2  Proof of Lemma 4

*Proof of Lemma 4.*  Suppose $\lambda$ is an eigenvalue of $\mathbf{T}_{\mathrm{NAG}}$, then we have

$$
\det(\mathbf{T}_{\mathrm{NAG}} - \lambda\mathbf{I}_{2mn}) = \det((\beta + \lambda^2 - (1+\beta)\lambda)\mathbf{I}_{mn} + (\eta(1+\beta)\lambda - \eta\beta)\mathbf{H}_0).
$$

Since $\mathbf{H}_0$ is symmetric, it can be simultaneously diagonalized with $\mathbf{I}$, hence the above equation becomes

$$
\lambda^2 - (1+\beta)\lambda + \beta + \eta(1+\beta)\lambda_i(\mathbf{H}_0)\lambda - \eta\beta\lambda_i(\mathbf{H}_0) = 0
$$

for some $1 \le i \le mn$. Solving the equation yields

$$
\lambda = \frac{1}{2}\left((1+\beta)(1 - \eta\lambda_i(\mathbf{H}_0)) \pm \sqrt{(1 - \eta\lambda_i(\mathbf{H}_0))\left(-4\beta + (1+\beta)^2(1 - \eta\lambda_i(\mathbf{H}_0))\right)}\right).
$$

For $i > nr$, $\lambda_i(\mathbf{H}_0) = 0$, hence $\lambda = 1$ or $\lambda = \beta$. The corresponding eigen subspaces are

$$
\mathcal{H}_1 = \left\{(\mathbf{u}^\top, \mathbf{v}^\top)^\top \mid \mathbf{u} = \mathbf{v} \in \ker(\mathbf{H}_0)\right\},
$$
$$
\mathcal{H}_\beta = \left\{(\mathbf{u}^\top, \mathbf{v}^\top)^\top \mid \mathbf{u} = \beta\mathbf{v} \in \ker(\mathbf{H}_0)\right\}.
$$

The dimensions are $\dim(\mathcal{H}_1) = \dim(\mathcal{H}_\beta) = (m-r)n$. It is easy to verify that whenever $0 < \beta < 1$,

$$
\mathcal{H}_1 \oplus \mathcal{H}_\beta = \ker(\mathcal{H}_0) \times \ker(\mathcal{H}_0).
$$

The complement space of $\mathcal{H}_1 \oplus \mathcal{H}_\beta$ corresponds to the eigen subspace for non-trivial eigenvalues. By checking the dimension and orthogonality, we have

$$
(\mathcal{H}_1 \oplus \mathcal{H}_\beta)^\perp = \mathcal{H} \times \mathcal{H}.
$$

For $i \leq nr$, the subspace is $\mathcal{H} \times \mathcal{H}$ and the contraction condition requires

$$0 < \eta < \frac{2(1+\beta)}{(1+2\beta)\sigma_1^2(\mathbf{X}_0)} = \frac{2(1+\beta)}{(1+2\beta)L}.$$

By checking the monotonicity of $|\lambda|$ with respect to $1 - \eta\lambda_i(\mathbf{H}_0) \in [1 - \eta L, 1 - \eta\mu]$, we have

$$|\lambda| \leq \max\left\{ \frac{1}{2}\left( (1+\beta)(1-\eta\mu) + \sqrt{(1-\eta\mu)\left(-4\beta + (1+\beta)^2(1-\eta\mu)\right)} \right),\right.$$
$$\left. \frac{1}{2}\left( -(1+\beta)(1-\eta L) + \sqrt{(1-\eta L)\left(-4\beta + (1+\beta)^2(1-\eta L)\right)} \right) \right\}.$$

If we choose step size $\eta = \frac{1}{L}$, momentum $\beta = \frac{\sqrt{L}-\sqrt{\mu}}{\sqrt{L}+\sqrt{\mu}}$, then we have $|\lambda| \leq 1 - \sqrt{\frac{\mu}{L}}$. $\qquad \square$

### C.3 Proof of Lemma 5

*Proof of Lemma 5.* According to Lemma 3,

$$\boldsymbol{\xi}_t = \boldsymbol{\zeta}_t + \boldsymbol{\iota}_t,$$
$$\boldsymbol{\zeta}_t = \mathrm{vec}(\mathbf{P}_t\mathbf{Q}_t^\top) + \beta\,\mathrm{vec}(\eta\mathbf{R}_{t-1}\mathbf{Y}_{t-1}\mathbf{Q}_{t-1}^\top + \eta\mathbf{P}_{t-1}\mathbf{X}_{t-1}^\top\mathbf{R}_{t-1} + \mathbf{P}_{t-1}\mathbf{Q}_{t-1}^\top)$$
$$\boldsymbol{\iota}_t = (1+\beta)\eta(\mathbf{H}_0 - \mathbf{H}_t)\mathbf{r}_t - \beta\eta(\mathbf{H}_0 - \mathbf{H}_{t-1})\mathbf{r}_{t-1}.$$

We first bound $\|\mathbf{P}_t\|_\mathrm{F}$ and $\|\mathbf{Q}_t\|_\mathrm{F}$. For every $0 \leq s \leq t$, we have

$$\begin{aligned}
\|\mathbf{R}_s\mathbf{Y}_s\|_\mathrm{F} &\leq \|\mathbf{Y}_s\|\,\|\mathbf{R}_s\|_\mathrm{F} \\
&\leq (\|\mathbf{Y}_0\| + \|\mathbf{Y}_s - \mathbf{Y}_0\|)\,\|\mathbf{R}_s\|_\mathrm{F} \\
&\leq (\|\mathbf{Y}_0\| + \|\mathbf{Y}_s - \mathbf{Y}_0\|_\mathrm{F})\,\|\mathbf{R}_s\|_\mathrm{F} \\
&\leq (\|\mathbf{Y}_0\| + C_2)\,\|\mathbf{R}_s\|_\mathrm{F}.
\end{aligned}$$

Similarly,

$$\left\|\mathbf{R}_s^\top\mathbf{X}_s\right\|_\mathrm{F} \leq (\|\mathbf{X}_0\| + C_2)\,\|\mathbf{R}_s\|_\mathrm{F}.$$

By assumption, we have

$$\|\mathbf{R}_s\|_\mathrm{F} \leq \left\|\begin{pmatrix}\mathbf{r}_s \\ \mathbf{r}_{s-1}\end{pmatrix}\right\| \leq C_1\theta^s\left\|\begin{pmatrix}\mathbf{r}_0 \\ \mathbf{r}_{-1}\end{pmatrix}\right\|.$$

As a result, the momentum terms can be bounded:

$$\begin{aligned}
\|\mathbf{P}_t\|_\mathrm{F} &= \left\|\eta\mathbf{R}_t\mathbf{Y}_t + \eta\sum_{s=1}^{t}\beta^{t-s+1}\mathbf{R}_s\mathbf{Y}_s\right\|_\mathrm{F} \\
&\leq \eta\,\|\mathbf{R}_t\mathbf{Y}_t\|_\mathrm{F} + \eta\sum_{s=1}^{t}\beta^{t-s+1}\,\|\mathbf{R}_s\mathbf{Y}_s\|_\mathrm{F} \\
&\leq \eta(\|\mathbf{Y}_0\| + C_2)\left(\|\mathbf{R}_t\|_\mathrm{F} + \sum_{s=1}^{t}\beta^{t-s+1}\,\|\mathbf{R}_s\|_\mathrm{F}\right) \\
&\leq \eta C_1(\|\mathbf{Y}_0\| + C_2)\left(\theta^t + \sum_{s=1}^{t}\beta^{t-s+1}\theta^s\right)\left\|\begin{pmatrix}\mathbf{r}_0 \\ \mathbf{r}_{-1}\end{pmatrix}\right\| \\
&\leq \eta C_1(\|\mathbf{Y}_0\| + C_2)\frac{1}{1-\theta}\cdot\theta^t\left\|\begin{pmatrix}\mathbf{r}_0 \\ \mathbf{r}_{-1}\end{pmatrix}\right\|, \qquad\qquad (20)
\end{aligned}$$

and

$$\|\mathbf{Q}_t\|_\mathrm{F} \leq \eta C_1(\|\mathbf{X}_0\| + C_2)\frac{1}{1-\theta}\cdot\theta^t\left\|\begin{pmatrix}\mathbf{r}_0 \\ \mathbf{r}_{-1}\end{pmatrix}\right\|, \qquad\qquad (21)$$

where we use $\beta \leq \theta^2 < \theta$ in the last steps.

Next, we bound $\|\boldsymbol{\zeta}_t\|$. Using the triangle inequality, we get

$$\|\boldsymbol{\zeta}_t\| \leq \left\|\mathbf{P}_t\mathbf{Q}_t^\top\right\|_{\mathrm{F}} + \beta\left\|\eta\mathbf{R}_{t-1}\mathbf{Y}_{t-1}\mathbf{Q}_{t-1}^\top + \eta\mathbf{P}_{t-1}\mathbf{X}_{t-1}^\top\mathbf{R}_{t-1} + \mathbf{P}_{t-1}\mathbf{Q}_{t-1}^\top\right\|_{\mathrm{F}}.$$

For the first term, we have

$$\left\|\mathbf{P}_t\mathbf{Q}_t^\top\right\|_{\mathrm{F}} \leq \|\mathbf{P}_t\|_{\mathrm{F}}\|\mathbf{Q}_t\|_{\mathrm{F}} \leq \frac{\eta^2 C_1^2(\|\mathbf{X}_0\| + C_2)(\|\mathbf{Y}_0\| + C_2)}{(1-\theta)^2}\theta^{2t}\left\|\begin{pmatrix}\mathbf{r}_0 \\ \mathbf{r}_{-1}\end{pmatrix}\right\|^2.$$

For the second term, we have

$$\beta\left\|\eta\mathbf{R}_{t-1}\mathbf{Y}_{t-1}\mathbf{Q}_{t-1}^\top + \eta\mathbf{P}_{t-1}\mathbf{X}_{t-1}^\top\mathbf{R}_{t-1} + \mathbf{P}_{t-1}\mathbf{Q}_{t-1}^\top\right\|_{\mathrm{F}}$$
$$\leq \beta\left(\eta\|\mathbf{R}_{t-1}\|_{\mathrm{F}}\left(\|\mathbf{Y}_{t-1}\|\|\mathbf{Q}_{t-1}\|_{\mathrm{F}} + \|\mathbf{X}_{t-1}\|\|\mathbf{P}_{t-1}\|_{\mathrm{F}}\right) + \|\mathbf{P}_{t-1}\|_{\mathrm{F}}\|\mathbf{Q}_{t-1}\|_{\mathrm{F}}\right)$$
$$\leq \frac{\eta^2 C_1^2(\|\mathbf{X}_0\| + C_2)(\|\mathbf{Y}_0\| + C_2)(3 - 2\theta)}{(1-\theta)^2}\theta^{2t}\left\|\begin{pmatrix}\mathbf{r}_0 \\ \mathbf{r}_{-1}\end{pmatrix}\right\|^2.$$

As a result, we have

$$\|\boldsymbol{\zeta}_t\| \leq C_3\theta^{2t}\left\|\begin{pmatrix}\mathbf{r}_0 \\ \mathbf{r}_{-1}\end{pmatrix}\right\|^2,$$

where $C_3 = \frac{\eta^2 C_1^2(\|\mathbf{X}_0\| + C_2)(\|\mathbf{Y}_0\| + C_2)(4 - 2\theta)}{(1-\theta)^2}$.

We then show upper bound for $\|\boldsymbol{\iota}_t\|$. Using the triangle inequality, we get

$$\|\boldsymbol{\iota}_t\| \leq (1 + \beta)\eta\|(\mathbf{H}_0 - \mathbf{H}_t)\mathbf{r}_t\| + \beta\eta\|(\mathbf{H}_0 - \mathbf{H}_{t-1})\mathbf{r}_{t-1}\|. \tag{22}$$

For any $s \leq t$, we have

$$\|(\mathbf{H}_0 - \mathbf{H}_s)\mathbf{r}_s\| = \left\|\mathbf{R}_s(\mathbf{Y}_0\mathbf{Y}_0^\top - \mathbf{Y}_s\mathbf{Y}_s^\top) + (\mathbf{X}_0\mathbf{X}_0^\top - \mathbf{X}_s\mathbf{X}_s^\top)\mathbf{R}_s\right\|_{\mathrm{F}}$$
$$\leq \left\|\mathbf{R}_s(\mathbf{Y}_0\mathbf{Y}_0^\top - \mathbf{Y}_s\mathbf{Y}_s^\top)\right\|_{\mathrm{F}} + \left\|(\mathbf{X}_0\mathbf{X}_0^\top - \mathbf{X}_s\mathbf{X}_s^\top)\mathbf{R}_s\right\|_{\mathrm{F}}$$
$$\leq \left\|\mathbf{Y}_0\mathbf{Y}_0^\top - \mathbf{Y}_s\mathbf{Y}_s^\top\right\|\|\mathbf{R}_s\|_{\mathrm{F}} + \left\|\mathbf{X}_0\mathbf{X}_0^\top - \mathbf{X}_s\mathbf{X}_s^\top\right\|\|\mathbf{R}_s\|_{\mathrm{F}}$$
$$\leq (2\|\mathbf{Y}_0\| + \|\mathbf{Y}_s - \mathbf{Y}_0\|_{\mathrm{F}})\|\mathbf{Y}_s - \mathbf{Y}_0\|_{\mathrm{F}}\|\mathbf{R}_s\|_{\mathrm{F}}$$
$$\quad + (2\|\mathbf{X}_0\| + \|\mathbf{X}_s - \mathbf{X}_0\|_{\mathrm{F}})\|\mathbf{X}_s - \mathbf{X}_0\|_{\mathrm{F}}\|\mathbf{R}_s\|_{\mathrm{F}}$$
$$\leq 2(\|\mathbf{X}_0\| + \|\mathbf{Y}_0\| + C_2)C_2\|\mathbf{R}_s\|_{\mathrm{F}}$$
$$\leq 2(\|\mathbf{X}_0\| + \|\mathbf{Y}_0\| + C_2)C_1 C_2\theta^s\left\|\begin{pmatrix}\mathbf{r}_0 \\ \mathbf{r}_{-1}\end{pmatrix}\right\|.$$

Plugging it into (22) yields

$$\|\boldsymbol{\iota}_t\| \leq 2(\|\mathbf{X}_0\| + \|\mathbf{Y}_0\| + C_2)C_1 C_2((1 + \beta)\eta\theta^t + \beta\eta\theta^{t-1})\left\|\begin{pmatrix}\mathbf{r}_0 \\ \mathbf{r}_{-1}\end{pmatrix}\right\|$$
$$\leq C_4\theta^t\left\|\begin{pmatrix}\mathbf{r}_0 \\ \mathbf{r}_{-1}\end{pmatrix}\right\|,$$

where $C_4 = 2\eta(\|\mathbf{X}_0\| + \|\mathbf{Y}_0\| + C_2)C_1 C_2(1 + 2\theta)$.

Finally, given (9) and (20), we have

$$\|\mathbf{X}_{t+1} - \mathbf{X}_0\|_{\mathrm{F}} \leq \sum_{s=0}^{t}\|\mathbf{P}_s\|_{\mathrm{F}} \leq \frac{\eta C_1(\|\mathbf{Y}_0\| + C_2)}{(1-\theta)^2}\left\|\begin{pmatrix}\mathbf{r}_0 \\ \mathbf{r}_{-1}\end{pmatrix}\right\| \leq C_2,$$

where the last inequality is from our assumption on $C_2$. Similarly, by (21), we have

$$\|\mathbf{Y}_{t+1} - \mathbf{Y}_0\|_{\mathrm{F}} \leq \sum_{s=0}^{t}\|\mathbf{Q}_s\|_{\mathrm{F}} \leq \frac{\eta C_1(\|\mathbf{X}_0\| + C_2)}{(1-\theta)^2}\left\|\begin{pmatrix}\mathbf{r}_0 \\ \mathbf{r}_{-1}\end{pmatrix}\right\| \leq C_2.$$

$\square$

## C.4 Proof of Theorem 2

*Proof of Theorem 2.* By initialization, we have $\|\mathbf{r}_0\| = \|\mathbf{r}_{-1}\| = \|\mathbf{A}\|_{\mathrm{F}}$. Let $C_1$ to $C_4$ be constants defined in Lemma 5. Define $\rho = 1 - \frac{\sqrt{\mu}}{\sqrt{L}}$, $\theta = 1 - \frac{\sqrt{\mu}}{2\sqrt{L}}$, $a_t = C_1 \left\| (\mathbf{r}_t^\top, \mathbf{r}_{t-1}^\top) \right\|$, and $b_t = C_1 \|\boldsymbol{\xi}_t\|$ for $t \geq 0$. It is easy to verify that $\beta \leq \theta^2 < \theta < 1$ and $\rho < \theta < 1$. By Proposition 3 and lemmas 1 and 4 we have

$$a_{t+1} \leq \rho \cdot a_t + b_t$$

for all $t \geq 0$. It remains to show that $b_t \leq \theta^t \cdot c_0$. For the initial step, $a_0 = \sqrt{2}C_1 \|\mathbf{A}\|_{\mathrm{F}}$, $b_0 = 0$. Let $C_1 = \frac{\mu p}{4\sqrt{2}\|\mathbf{A}\|_{\mathrm{F}}(1+p)}$ and $C_2 = p\sqrt{L}$ where $p = \frac{\sqrt{\mu}}{144\sqrt{L}} \leq \frac{1}{144} < 1$, then we have

$$C_3 = \frac{\mu p^3 (2 + \sqrt{\frac{\mu}{L}})}{8 \|\mathbf{A}\|_{\mathrm{F}}^2 (1+p)}, \quad C_4 = \frac{\mu p^2 (3 - \sqrt{\frac{\mu}{L}})}{2\sqrt{2} \|\mathbf{A}\|_{\mathrm{F}}}.$$

Let $c_0 = \sqrt{2}C_1(\sqrt{2}C_3 \|\mathbf{A}\|_{\mathrm{F}} + C_4) \|\mathbf{A}\|_{\mathrm{F}}$, then we can show the following relations:

$$a_0 + \frac{c_0}{\theta - \rho} \leq \sqrt{2}C_1^2 \|\mathbf{A}\|_{\mathrm{F}} \quad \text{and} \quad C_1 \geq 1. \tag{23}$$

Indeed, by Proposition 1, with probability at least $1 - \delta$, our choice of $c$ guarantees

$$\mu = \sigma_r^2(\mathbf{X}_0) \geq \frac{\tau^2(\sqrt{d} - \sqrt{r-1})^2 c^2 \sigma_r^2(\mathbf{A})}{d} \geq \frac{4\sqrt{2} \|\mathbf{A}\|_{\mathrm{F}} (1+p)}{p}, \tag{24}$$

thus $C_1 \geq 1$. Here, we use the bound $p \leq \frac{1}{144} < 1$ to verify the numerical constant. It remains to show

$$a_0 + \frac{c_0}{\theta - \rho} \leq \sqrt{2}C_1^2 \|\mathbf{A}\|_{\mathrm{F}},$$

which is equivalent to

$$\|\mathbf{A}\|_{\mathrm{F}} + \frac{p^3 \sqrt{\mu L}(2 + \sqrt{\frac{\mu}{L}})}{2\sqrt{2}(1+p)} + \frac{p^2 \sqrt{\mu L}(3 - \sqrt{\frac{\mu}{L}})}{\sqrt{2}} \leq \frac{\mu p}{4\sqrt{2}(1+p)},$$

Since we set $p = \frac{\sqrt{\mu}}{144\sqrt{L}} < 1$, each one of the three terms on the left hand side is upper bounded by $\frac{\mu p}{12\sqrt{2}(1+p)}$, hence the inequality holds. The relations (23) guarantee the induction conditions in Lemma 5, thus we have

$$\|\mathbf{r}_{t+1}\| \leq \sqrt{2}C_1 \theta^{t+1} \|\mathbf{A}\|_{\mathrm{F}} \leq \frac{c^2 \sigma_1^2(\mathbf{A})}{64 \|\mathbf{A}\|_{\mathrm{F}} \mathrm{cond}(\mathbf{X}_0)} \theta^{t+1} \|\mathbf{A}\|_{\mathrm{F}},$$

where the last inequality uses $p > 0$ and Proposition 1. $\qquad\square$

# D   Missing Proofs for NAG in Section 4

Let $\tilde{\mathbf{r}}_t = \mathrm{vec}(\tilde{\mathbf{R}}_t)$, then we have the following dynamics.

**Lemma 7.** *Let $\mathbf{P}_t = \mathbf{X}_{t+1} - \mathbf{X}_t$ and $\mathbf{Q}_t = \mathbf{Y}_{t+1} - \mathbf{Y}_t$ denote the momentum. Let $\mathbf{R}_t = \mathbf{X}_t \mathbf{Y}_t^\top \mathbf{D} - \mathbf{L}$ denote the residual, $\tilde{\mathbf{R}}_t = \mathbf{X}_t \mathbf{Y}_t^\top \mathbf{D} \mathbf{D}^\top - \mathbf{L} \mathbf{D}^\top$ denote the projected residual, $\tilde{\mathbf{r}}_t = \mathrm{vec}(\tilde{\mathbf{R}}_t) \in \mathbb{R}^{mn}$. Then NAG has the following dynamics:*

$$\begin{pmatrix} \tilde{\mathbf{r}}_{t+1} \\ \tilde{\mathbf{r}}_t \end{pmatrix} = \begin{pmatrix} (1+\beta)(\mathbf{I}_{mn} - \eta\mathbf{H}_0) & -\beta(\mathbf{I}_{mn} - \eta\mathbf{H}_0) \\ \mathbf{I}_{mn} & 0 \end{pmatrix} \begin{pmatrix} \tilde{\mathbf{r}}_t \\ \tilde{\mathbf{r}}_{t-1} \end{pmatrix} + \begin{pmatrix} \boldsymbol{\xi}_t \\ 0 \end{pmatrix}, \tag{25}$$

*where*

$$\begin{aligned}
\mathbf{H}_t &= (\mathbf{D}\mathbf{D}^\top \mathbf{Y}_t \mathbf{Y}_t^\top) \otimes \mathbf{I}_m + (\mathbf{D}\mathbf{D}^\top) \otimes (\mathbf{X}_t \mathbf{X}_t^\top), \\
\boldsymbol{\xi}_t &= \boldsymbol{\zeta}_t + \boldsymbol{\iota}_t, \\
\boldsymbol{\zeta}_t &= \mathrm{vec}(\mathbf{P}_t \mathbf{Q}_t^\top \mathbf{D} \mathbf{D}^\top) + \beta \, \mathrm{vec}(\mathbf{P}_{t-1} \mathbf{Q}_{t-1}^\top \mathbf{D} \mathbf{D}^\top) \\
&\quad + \beta\eta \, \mathrm{vec}((\tilde{\mathbf{R}}_{t-1} \mathbf{Y}_{t-1} \mathbf{Q}_{t-1}^\top + \mathbf{P}_{t-1} \mathbf{X}_{t-1}^\top \tilde{\mathbf{R}}_{t-1}) \mathbf{D} \mathbf{D}^\top), \\
\boldsymbol{\iota}_t &= (1+\beta)\eta(\mathbf{H}_0 - \mathbf{H}_t)\tilde{\mathbf{r}}_t - \beta\eta(\mathbf{H}_0 - \mathbf{H}_{t-1})\tilde{\mathbf{r}}_{t-1}.
\end{aligned}$$

*Proof of Lemma 7.* We denote $\mathbf{R}_t = \mathbf{X}_t \mathbf{Y}_t^\top \mathbf{D} - \mathbf{L}$ as the residual, $\tilde{\mathbf{R}}_t = \mathbf{R}_t \mathbf{D}^\top$ as the projected residual, then the NAG update for (10) can be written as

$$\begin{pmatrix} \mathbf{X}_{t+1} \\ \mathbf{Y}_{t+1} \end{pmatrix} = \begin{pmatrix} (1+\beta)(\mathbf{X}_t - \eta\tilde{\mathbf{R}}_t \mathbf{Y}_t) - \beta(\mathbf{X}_{t-1} - \eta\tilde{\mathbf{R}}_{t-1}\mathbf{Y}_{t-1}) \\ (1+\beta)(\mathbf{Y}_t - \eta\tilde{\mathbf{R}}_t^\top \mathbf{X}_t) - \beta(\mathbf{Y}_{t-1} - \eta\tilde{\mathbf{R}}_{t-1}^\top \mathbf{X}_{t-1}) \end{pmatrix}. \tag{26}$$

The result follows from (26) by direct computation. $\square$

**Lemma 8.** *Let $\mathcal{H} \subseteq \mathbb{R}^{mn}$ denote the linear subspace containing all eigenvectors of $\mathbf{H}_0 = (\mathbf{D}\mathbf{D}^\top) \otimes (\mathbf{X}_0 \mathbf{X}_0^\top)$ with positive eigenvalues. If $\mathrm{col}(\mathbf{X}_0) = \mathrm{col}(\mathbf{L})$ and $\mathbf{Y}_0 = 0$, then we have*

$$\mathcal{H} = \mathrm{col}(\mathbf{D} \otimes \mathbf{L}) \quad and \quad \{\tilde{\mathbf{r}}_t, \boldsymbol{\xi}_t\}_{t \geq 0} \subset \mathcal{H},$$

*where $\mathbf{H}_0$, $\tilde{\mathbf{r}}_t$ and $\boldsymbol{\xi}_t$ are defined as in Lemma 7.*

*Proof.* By Theorem 4.2.15 in Horn and Johnson [1994], we have the following eigenvalue decomposition for Kronecker product:

$$\mathbf{H}_0 = (\mathbf{U}_D \otimes \mathbf{U}_0)(\boldsymbol{\Sigma}_D^2 \otimes \boldsymbol{\Sigma}_0^2)(\mathbf{U}_D \otimes \mathbf{U}_0)^\top,$$

where $\mathbf{D} = \mathbf{U}_D \boldsymbol{\Sigma}_D \mathbf{V}_D^\top$ and $\mathbf{X}_0 = \mathbf{U}_0 \boldsymbol{\Sigma}_0 \mathbf{V}_0^\top$ are singular value decompositions of $\mathbf{D}$ and $\mathbf{X}_0$. Therefore, we have

$$\mathcal{H} = \mathrm{col}(\mathbf{U}_D \otimes \mathbf{U}_0) = \mathrm{col}(\mathbf{D} \otimes \mathbf{X}_0) = \mathrm{col}(\mathbf{D} \otimes \mathbf{L}).$$

In particular, the eigenvalues (not ordered) are

$$\lambda_{(i-1)m+j}(\mathbf{H}_0) = \lambda_i(\mathbf{D}\mathbf{D}^\top)\lambda_j(\mathbf{X}_0 \mathbf{X}_0^\top) = \sigma_i^2(\mathbf{D})\sigma_j^2(\mathbf{X}_0), \ i \in [n], j \in [m],$$

where $\sigma_j(\mathbf{X}_0) > 0$ for $1 \leq j \leq r$, $\sigma_j(\mathbf{X}_0) = 0$ for $r+1 \leq j \leq d$. By Assumption 1, $\mathbf{L} = \mathbf{A}\mathbf{D}$, thus we have

$$\mathrm{vec}(\mathbf{L}\mathbf{D}^\top) = \mathrm{vec}(\mathbf{L}\mathbf{I}_N \mathbf{D}^\top) = (\mathbf{D} \otimes \mathbf{L})\mathbf{I}_N \in \mathrm{col}(\mathbf{D} \otimes \mathbf{L}) = \mathcal{H}.$$

Meanwhile,

$$\mathrm{vec}(\mathbf{X}_t \mathbf{Y}_t^\top \mathbf{D}\mathbf{D}^\top) = (\mathbf{D} \otimes \mathbf{X}_t)\,\mathrm{vec}(\mathbf{Y}_t^\top \mathbf{D}) \in \mathrm{col}(\mathbf{D} \otimes \mathbf{X}_t) \subseteq \mathrm{col}(\mathbf{D} \otimes \mathbf{X}_0) = \mathcal{H},$$

thus we have $\tilde{\mathbf{r}}_t \in \mathcal{H}$. Similarly, we have $\boldsymbol{\xi}_t \in \mathcal{H}$. $\square$

**Lemma 9** (NAG contraction). *If we choose step size $\eta = \frac{1}{\tilde{L}}$ and momentum $\beta = \frac{\sqrt{\tilde{L}}-\sqrt{\tilde{\mu}}}{\sqrt{\tilde{L}}+\sqrt{\tilde{\mu}}}$ where $\tilde{L} = \sigma_1^2(\mathbf{X}_0) \cdot \lambda_{\max}(\mathbf{D}\mathbf{D}^\top)$, $\tilde{\mu} = \sigma_r^2(\mathbf{X}_0) \cdot \lambda_{\min}(\mathbf{D}\mathbf{D}^\top)$, then for all $(\mathbf{u}, \mathbf{v}) \in \mathcal{H} \times \mathcal{H}$, $\mathcal{H}$ defined in Lemma 8,*

$$\left\| \mathbf{T}_{\mathrm{NAG}} \begin{pmatrix} \mathbf{u} \\ \mathbf{v} \end{pmatrix} \right\| \leq \left(1 - \sqrt{\frac{\tilde{\mu}}{\tilde{L}}}\right) \left\| \begin{pmatrix} \mathbf{u} \\ \mathbf{v} \end{pmatrix} \right\|.$$

*Proof.* Following the same line of proof for Lemma 4 in Appendix C.2 and substituting the eigenvalues in Lemma 8, we obtain the result. $\square$

**Lemma 10.** *Suppose $0 < \beta \leq \theta^2 < \theta < 1$. If there exist some constants $C_1$ and $C_2$ such that for any $s \leq t$, the NAG dynamics (7) yields $\left\| \begin{pmatrix} \tilde{\mathbf{r}}_s \\ \tilde{\mathbf{r}}_{s-1} \end{pmatrix} \right\| \leq C_1 \theta^s \left\| \begin{pmatrix} \tilde{\mathbf{r}}_0 \\ \tilde{\mathbf{r}}_{-1} \end{pmatrix} \right\|$, $\|\mathbf{X}_s - \mathbf{X}_0\|_{\mathrm{F}} \leq C_2$, and $\|\mathbf{Y}_s - \mathbf{Y}_0\|_{\mathrm{F}} \leq C_2$, then we have*

$$\|\boldsymbol{\zeta}_t\| \leq C_3 \theta^{2t} \left\| \begin{pmatrix} \tilde{\mathbf{r}}_0 \\ \tilde{\mathbf{r}}_{-1} \end{pmatrix} \right\|^2, \quad and \quad \|\boldsymbol{\iota}_t\| \leq C_4 \theta^t \left\| \begin{pmatrix} \tilde{\mathbf{r}}_0 \\ \tilde{\mathbf{r}}_{-1} \end{pmatrix} \right\|$$

*for some constants $C_3$ and $C_4$ depending on $C_1$ and $C_2$. Moreover, if $C_1$ and $C_2$ satisfy*

$$(\max(\|\mathbf{X}_0\|, \|\mathbf{Y}_0\|) + C_2)\eta C_1 \left\| \begin{pmatrix} \tilde{\mathbf{r}}_0 \\ \tilde{\mathbf{r}}_{-1} \end{pmatrix} \right\| \leq (1-\theta)^2 C_2,$$

*then we have*

$$\|\mathbf{X}_{t+1} - \mathbf{X}_0\|_{\mathrm{F}} \leq C_2, \quad \|\mathbf{Y}_{t+1} - \mathbf{Y}_0\|_{\mathrm{F}} \leq C_2.$$

*Proof of Lemma 10.* Following the same line of proof for Lemma 5 in Appendix C.3, we have

$$\|\mathbf{P}_t\|_{\mathrm{F}} \le \eta C_1(\|\mathbf{Y}_0\| + C_2)\frac{1}{1-\theta} \cdot \theta^t \left\| \begin{pmatrix} \tilde{\mathbf{r}}_0 \\ \tilde{\mathbf{r}}_{-1} \end{pmatrix} \right\|, \tag{27}$$

and

$$\|\mathbf{Q}_t\|_{\mathrm{F}} \le \eta C_1(\|\mathbf{X}_0\| + C_2)\frac{1}{1-\theta} \cdot \theta^t \left\| \begin{pmatrix} \tilde{\mathbf{r}}_0 \\ \tilde{\mathbf{r}}_{-1} \end{pmatrix} \right\|. \tag{28}$$

As a result, we have

$$\left\|\mathbf{P}_t\mathbf{Q}_t^\top \mathbf{D}\mathbf{D}^\top\right\|_{\mathrm{F}} \le \lambda_1(\mathbf{D}\mathbf{D}^\top)\|\mathbf{P}_t\|_{\mathrm{F}}\|\mathbf{Q}_t\|_{\mathrm{F}} \le \frac{\eta^2 C_1^2(\|\mathbf{X}_0\| + C_2)(\|\mathbf{Y}_0\| + C_2)\lambda_1(\mathbf{D}\mathbf{D}^\top)}{(1-\theta)^2}\theta^{2t}\left\| \begin{pmatrix} \tilde{\mathbf{r}}_0 \\ \tilde{\mathbf{r}}_{-1} \end{pmatrix} \right\|^2,$$

and

$$\beta \left\|(\eta\tilde{\mathbf{R}}_{t-1}\mathbf{Y}_{t-1}\mathbf{Q}_{t-1}^\top + \eta\mathbf{P}_{t-1}\mathbf{X}_{t-1}^\top\tilde{\mathbf{R}}_{t-1} + \mathbf{P}_{t-1}\mathbf{Q}_{t-1}^\top)\mathbf{D}\mathbf{D}^\top\right\|_{\mathrm{F}}$$
$$\le \beta\lambda_1(\mathbf{D}\mathbf{D}^\top)\left(\eta\left\|\tilde{\mathbf{R}}_{t-1}\right\|_{\mathrm{F}}(\|\mathbf{Y}_{t-1}\|\|\mathbf{Q}_{t-1}\|_{\mathrm{F}} + \|\mathbf{X}_{t-1}\|\|\mathbf{P}_{t-1}\|_{\mathrm{F}}) + \|\mathbf{P}_{t-1}\|_{\mathrm{F}}\|\mathbf{Q}_{t-1}\|_{\mathrm{F}}\right)$$
$$\le \frac{\eta^2 C_1^2(\|\mathbf{X}_0\| + C_2)(\|\mathbf{Y}_0\| + C_2)(3-2\theta)\lambda_1(\mathbf{D}\mathbf{D}^\top)}{(1-\theta)^2}\theta^{2t}\left\| \begin{pmatrix} \tilde{\mathbf{r}}_0 \\ \tilde{\mathbf{r}}_{-1} \end{pmatrix} \right\|^2,$$

Combining the inequalities, we get

$$\|\boldsymbol{\zeta}_t\| \le C_3\theta^{2t}\left\| \begin{pmatrix} \tilde{\mathbf{r}}_0 \\ \tilde{\mathbf{r}}_{-1} \end{pmatrix} \right\|^2,$$

where $C_3 = \frac{\eta^2 C_1^2(\|\mathbf{X}_0\|+C_2)(\|\mathbf{Y}_0\|+C_2)(4-2\theta)\lambda_1(\mathbf{D}\mathbf{D}^\top)}{(1-\theta)^2}$.

Similarly, we have

$$\|\boldsymbol{\iota}_t\| \le 2(\|\mathbf{X}_0\| + \|\mathbf{Y}_0\| + C_2)C_1C_2\lambda_1(\mathbf{D}\mathbf{D}^\top)((1+\beta)\eta\theta^t + \beta\eta\theta^{t-1})\left\| \begin{pmatrix} \tilde{\mathbf{r}}_0 \\ \tilde{\mathbf{r}}_{-1} \end{pmatrix} \right\|$$
$$\le C_4\theta^t\left\| \begin{pmatrix} \tilde{\mathbf{r}}_0 \\ \tilde{\mathbf{r}}_{-1} \end{pmatrix} \right\|,$$

where $C_4 = 2\eta(\|\mathbf{X}_0\| + \|\mathbf{Y}_0\| + C_2)C_1C_2(1+2\theta)\lambda_1(\mathbf{D}\mathbf{D}^\top)$.

Finally, by (27), we have

$$\|\mathbf{X}_{t+1} - \mathbf{X}_0\|_{\mathrm{F}} \le \sum_{s=0}^t \|\mathbf{P}_s\|_{\mathrm{F}} \le \frac{\eta C_1(\|\mathbf{Y}_0\| + C_2)}{(1-\theta)^2}\left\| \begin{pmatrix} \tilde{\mathbf{r}}_0 \\ \tilde{\mathbf{r}}_{-1} \end{pmatrix} \right\| \le C_2,$$

where the last inequality is from our assumption on $C_2$. Similarly, by (28), we have

$$\|\mathbf{Y}_{t+1} - \mathbf{Y}_0\|_{\mathrm{F}} \le \sum_{s=0}^t \|\mathbf{Q}_s\|_{\mathrm{F}} \le \frac{\eta C_1(\|\mathbf{X}_0\| + C_2)}{(1-\theta)^2}\left\| \begin{pmatrix} \tilde{\mathbf{r}}_0 \\ \tilde{\mathbf{r}}_{-1} \end{pmatrix} \right\| \le C_2.$$

$\square$

## D.1 Proof of Theorem 3

*Proof of Theorem 3.* By initialization, we have $\|\tilde{\mathbf{r}}_0\| = \|\tilde{\mathbf{r}}_{-1}\| = \|\mathbf{L}\mathbf{D}^\top\|_{\mathrm{F}}$. Let $C_1$ to $C_4$ be constants defined in Lemma 10. Define $\rho = 1 - \frac{\sqrt{\tilde{\mu}}}{\sqrt{\tilde{L}}}$, $\theta = 1 - \frac{\sqrt{\tilde{\mu}}}{2\sqrt{\tilde{L}}}$, $a_t = C_1\left\|\begin{pmatrix} \tilde{\mathbf{R}}_t & \tilde{\mathbf{R}}_{-1} \end{pmatrix}\right\|_{\mathrm{F}}$, and $b_t = C_1\|\boldsymbol{\xi}_t\|$ for $t \ge 0$. It is easy to verify that $\beta \le \theta^2 < \theta < 1$ and $\rho < \theta < 1$. By Lemmas 7 to 9 we have

$$a_{t+1} \le \rho \cdot a_t + b_t$$

for all $t \geq 0$. It remains to show that $b_t \leq \theta^t \cdot c_0$. For the initial step, $a_0 = \sqrt{2}C_1 \left\| \mathbf{LD}^\top \right\|_\mathrm{F}$, $b_0 = 0$. Let $C_1 = \frac{\tilde{\mu}p}{4\sqrt{2}\|\mathbf{LD}^\top\|_\mathrm{F}(1+p)}$ and $C_2 = p\sqrt{L}$ where $p = \frac{\sqrt{\tilde{\mu}}}{144\sqrt{\tilde{L}}} \leq \frac{1}{144} < 1$, then we have

$$C_3 = \frac{\tilde{\mu}p^3}{8\left\| \mathbf{LD}^\top \right\|_\mathrm{F}^2 (1+p)} \left( 2 + \sqrt{\frac{\tilde{\mu}}{\tilde{L}}} \right), \quad C_4 = \frac{\tilde{\mu}p^2}{2\sqrt{2}\left\| \mathbf{LD}^\top \right\|_\mathrm{F}} \left( 3 - \sqrt{\frac{\tilde{\mu}}{\tilde{L}}} \right).$$

Let $c_0 = \sqrt{2}C_1(\sqrt{2}C_3 \left\| \mathbf{LD}^\top \right\|_\mathrm{F} + C_4) \left\| \mathbf{LD}^\top \right\|_\mathrm{F}$, then we can show the following relations: Given our choice of constants, there hold

$$a_0 + \frac{c_0}{\theta - \rho} \leq \sqrt{2}C_1^2 \left\| \mathbf{A} \right\|_\mathrm{F} \quad \text{and} \quad C_1 \geq 1. \tag{29}$$

Indeed, by (11), we have $C_1 \geq 1$. It remains to show

$$a_0 + \frac{c_0}{\theta - \rho} \leq \sqrt{2}C_1^2 \left\| \mathbf{LD}^\top \right\|_\mathrm{F},$$

which is equivalent to

$$\left\| \mathbf{LD}^\top \right\|_\mathrm{F} + \frac{\sqrt{\tilde{\mu}\tilde{L}}p^3}{2\sqrt{2}(1+p)} \left( 2 + \sqrt{\frac{\tilde{\mu}}{\tilde{L}}} \right) + \frac{\sqrt{\tilde{\mu}\tilde{L}}p^2}{\sqrt{2}} \left( 3 - \sqrt{\frac{\tilde{\mu}}{\tilde{L}}} \right) \leq \frac{\tilde{\mu}p}{4\sqrt{2}(1+p)}.$$

By (11) and $p = \frac{\sqrt{\tilde{\mu}}}{144\sqrt{\tilde{L}}} < 1$, each one of the three terms on the left hand side is upper bounded by $\frac{\tilde{\mu}p}{12\sqrt{2}(1+p)}$, hence the inequality holds. (29) guarantees the induction conditions in Lemma 10, thus we have

$$\|\tilde{\mathbf{r}}_{t+1}\| \leq \sqrt{2}C_1\theta^{t+1} \left\| \mathbf{LD}^\top \right\|_\mathrm{F} \leq \frac{\tilde{\mu}}{576 \left\| \mathbf{LD}^\top \right\|_\mathrm{F}} \left( 1 - \frac{\sqrt{\tilde{\mu}}}{2\sqrt{\tilde{L}}} \right)^{t+1} \left\| \mathbf{LD}^\top \right\|_\mathrm{F}.$$

By Assumption 1, we have $\mathrm{row}(\mathbf{L}) \in \mathrm{row}(\mathbf{D}) = \mathrm{col}(\mathbf{D}^\top)$, thus we have

$$\begin{aligned}
\|\mathbf{R}_t\|_\mathrm{F} &= \left\| \mathbf{X}_t\mathbf{Y}_t^\top\mathbf{D} - \mathbf{L} \right\|_\mathrm{F} \\
&\leq \sigma_{\min}^{-1}(\mathbf{D}) \left\| (\mathbf{X}_t\mathbf{Y}_t^\top\mathbf{D} - \mathbf{L})\mathbf{D}^\top \right\|_\mathrm{F} \\
&\leq \frac{\sigma_r^2(\mathbf{X}_0)\sigma_{\min}(\mathbf{D})}{576} \left( 1 - \frac{\sigma_r(\mathbf{X}_0)\sqrt{\lambda_{\min}(\mathbf{DD}^\top)}}{2\sigma_1(\mathbf{X}_0)\sqrt{\lambda_{\max}(\mathbf{DD}^\top)}} \right)^t.
\end{aligned}$$

$\square$

## D.2 Proof of Corollaries

*Proof of Corollary 1.* By Proposition 1, $\mathrm{cond}(\mathbf{X}_0) = O(\frac{d \cdot \mathrm{cond}(\mathbf{L})}{\tau(d-r+1)})$ with probability at least $1 - \delta$, where $\delta = 3e^{-\min\{(d-r+1)\log\frac{1}{c_1\tau}, c_2d, \frac{d}{2}\}}$. Plugging it in Theorem 3 yields the result. $\square$

*Proof of Corollary 2.* After orthonormalization, we have $\mathrm{cond}(\mathbf{X}_0) = 1$. The result follows immediately from Theorem 3. $\square$

*Proof of Corollary 3.* By Propositions 4 and 5, $\mathrm{cond}(\mathbf{X}_0) = O(\frac{d}{\tau(d-m+1)})$ with probability at least $1 - \delta$, where $\delta = 3e^{-\min\{(d-m+1)\log\frac{1}{c_1\tau}, c_2d, \frac{d}{2}\}}$. Plugging it in Theorem 3 yields the result. $\square$

## E   Additional Experiments

This section provides additional experiments. Firstly, we investigate larger-sized problems by setting $(m, n) = (1200, 1000)$ for matrix factorization and $(m, n, N) = (500, 400, 600)$ for linear neural networks. We keep other settings the same as for Figure 2 and compare the performances of GD and NAG. The results are provided in Figure 4. As illustrated, the conclusion that NAG performs better than GD and overparameterization accelerates convergence remains valid for large matrices.

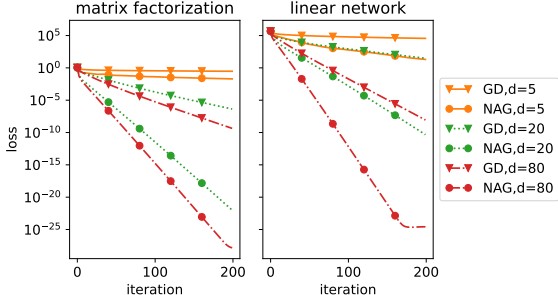

Figure 4: *GD and NAG on large matrices exhibit similar behavior to small matrices in Figure 2. Left: matrix factorization with $m = 1200$ and $n = 1000$. Right: linear neural networks with $m = 500$, $n = 400$, $N = 600$.*

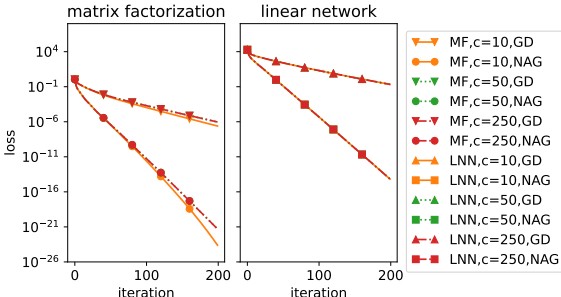

Figure 5: *GD and NAG with different values of c. When c is sufficiently large, changing its value would not significantly affect the convergence rate.*

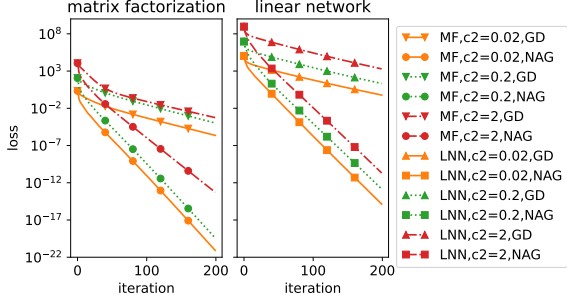

Figure 6: *GD and NAG with initialization $\mathbf{X}_0 = c_1 \mathbf{A} \mathbf{\Phi}_1$, $\mathbf{Y}_0 = c_2 \mathbf{\Phi}_2$, $c_1 = 50$. The initial loss (intercept) increases as $c_2$ increases within a range, while the convergence rate (slope) does not change significantly.*

Secondly, we conduct additional experiments on GD and NAG with different values of $c$ and plot the results in Figure 5. We set $d = 20$, while other settings remain the same as in Figure 2. As illustrated, when $c$ is sufficiently large, further increasing $c$ has little effect on the convergence rate, which is consistent with our theory.

We also investigate general unbalanced initialization $\mathbf{X}_0 = c_1 \mathbf{A} \mathbf{\Phi}_1 \in \mathbb{R}^{m \times d}$, $\mathbf{Y}_0 = c_2 \mathbf{\Phi}_2 \in \mathbb{R}^{n \times d}$, where $[\mathbf{\Phi}_1]_{i,j} \sim \mathcal{N}(0, 1/d)$ and $[\mathbf{\Phi}_2]_{i,j} \sim \mathcal{N}(0, 1/n)$. We set $d = 20$, while other settings remain the same as in Figure 2. We keep $c_1 = 50$ and set different values of $c_2$. The results are plotted in Figure 6. As illustrated, changing $c_2$ within a range only results in different initial losses (intercept), while the convergence rates (slope) are not significantly affected. This supports our claim in Remark 1.

