# OpenReview forum: "Provable Acceleration of Nesterov's Accelerated Gradient for Asymmetric Matrix Factorization and Linear Neural Networks"
_NeurIPS.cc/2024/Conference — NeurIPS 2024 poster_

### Official Review · Reviewer_u7PK · 2024-06-26

**Soundness:** 3
**Presentation:** 3
**Contribution:** 3
**Rating:** 7
**Confidence:** 4

**Summary:**

In this study, the authors established the convergence rate of the gradient descent and Nesterov's accelerated gradient descent methods for the asymmetric matrix factorization and the 2-layer linear network. The authors proved that an unbalanced initialization can lead to linear convergence of both methods, and the Nesterov's acceleration result in a faster convergence rate. Numerical experiments are implemented to support the theory.

**Strengths:**

This paper is clearly written and easy to follow in general. The results are novel and inspiring to audiences in the non-convex optimization field. I did not check the proofs in the appendix due to time limit, but the part in the main manuscript looks correct to me.

**Weaknesses:**

I do not see major weaknesses. The authors could consider discussing more related literature and include more intermediate steps in the main manuscript. Also, I think the authors can consider illustrating the effect of an unbalanced initialization in numerical experiments.

**Questions:**

- I think the following two papers also discussed the asymmetric low-rank matrix optimization problem. It would be better if the authors could discuss them and the references therein:

[1] Zhang, H., Bi, Y., & Lavaei, J. (2021). General low-rank matrix optimization: Geometric analysis and sharper bounds. Advances in Neural Information Processing Systems, 34, 27369-27380.

[2] Bi, Y., Zhang, H., & Lavaei, J. (2022, June). Local and global linear convergence of general low-rank matrix recovery problems. In Proceedings of the AAAI Conference on Artificial Intelligence (Vol. 36, No. 9, pp. 10129-10137).


- Following the above comment, I wonder if the authors can briefly discuss the potential and challenges of extending the results to more general low-rank matrix optimization problems.

- Lines 79-80: I am not sure why the method in [Stöger and Soltanolkotabi, 2021] is considered a preconditioned method.

- Theorem 1: In my understanding, the value $\|R_t\|_F$ should be proportional to the scale of $A$. However, in the bound of $\|R_t\|_F$, the right hand-side grows with the size of $A$. Maybe there is a typo?

- The above comment also applies to Theorem 2.

- It would be better to mention in Theorems 1-2 that the bound $T$ is derived using the bound in Proposition 1?

- Line 182: I think the shrinkage rate $\theta \in (0, \rho]$?

- I wonder if the authors can compare the performance of GD/NAG with different values of c?

**Limitations:**

See my comments in the Questions section.

---

> ### Author Rebuttal · Authors · 2024-08-07
>
> Thank you very much for your positive feedback and constructive comments. Our responses to each of the comments are listed below.
>
> > Q1: I think the following two papers also discussed the asymmetric low-rank matrix optimization problem.
>
> We thank the reviewer for pointing out these related works, and we will cite [1] and [2] and discuss them in the next version.
>
> > Q2: I wonder if the authors can briefly discuss the potential and challenges of extending the results to more general low-rank matrix optimization problems.
>
> Generalizing the results to the general low-rank matrix optimization problem $\min_{X,Y}f(XY^\top)$ is challenging. One of the challenges is the possible non-linearity of the gradient w.r.t. $X$. In our proof, a key step is to show the residual and the error terms in all iterations are in the contraction subspace of the dynamics, which requires the gradient of $f$ to preserve the column space of $X$. Unfortunately, such a property does not hold for general loss functions.
> Nevertheless, generalization to neural networks with non-linear activations, $\min_{X,Y}|L-X\sigma(Y^\top D)|_F^2$, will not have this issue, and we believe it is an interesting topic that is worth future investigation.
>
> > Q3: Lines 79-80: I am not sure why the method in [Stöger and Soltanolkotabi, 2021] is considered a preconditioned method.
>
> Thank you for pointing out this misplacement. The method in [Stöger and Soltanolkotabi, 2021] is not a preconditioned one. We cite it to support the former claim " Overparameterization may heavily slow down convergence", as it shows $O(\kappa^2\log\frac{1}{\epsilon})$ for exact parameterization case and $O(\kappa^6\log\frac{\kappa}{\epsilon})$ for overparameterization case.
> To avoid confusion, we will move it before the comma in the next version.
>
> >  Q4 & 5: Theorem 1 & 2: In my understanding, the value $|R_t|_F$ should be proportional to the scale of $A$. However, in the bound of $|R_t|_F$, the right hand-side grows with the size of $A$. Maybe there is a typo?
>
> There is no typo here. The bound on $|R_t|_F$ implicitly depends on $|A|_F$ through the choice of $c$. The Theorems require $c^2\geq\underline{c}^2=O(|A|_F)$, and the RHS grows with $c^2$, so $|R_t|_F$ implicitly depends on $|A|_F$. When $c$ is larger than $\underline{c}$, $|A|_F$ is dominated by $c^2$ and our results still hold. When $c$ is too small, our proof does not work.
>
> Moreover, the RHS does not explicitly depend on the size of $A$. The size might affect the bound through $|A|_F$ and $\sigma_i(A)$, but there is no explicit dependence on the dimensions $m$ and $n$. The bound only depends on the rank $r$ and the overparameterization $d$, both are not directly related to the size of A.
>
> > Q6: It would be better to mention in Theorems 1-2 that the bound $T$ is derived using the bound in Proposition 1
>
> Thank you for the suggestion. We will mention the use of quantitative results in Proposition 1 in the next version.
>
> > Q7: Line 182: I think the shrinkage rate $\theta\in(0,\rho]$?
>
> It is not a typo, the error shrinkage rate $\theta$ is larger than the "ideal shrinkage rate"  $\rho$ given by the condition number of the linear part of the dynamics. Intuitively, the existence of nonlinear error terms will slow down convergence, so $|R_t|$ will shrink at a rate $\theta$ slower than $\rho$, namely $\theta>\rho$. Then by Lemma 3, all error terms can be controlled by sequences with shrinkage rate $\theta$. The auxiliary Lemma 6 provided in Appendix B.1 then guarantees that the final residual $|R_t|_F$ is controlled by $O(\theta^t)$.
>
> > Q8: I wonder if the authors can compare the performance of GD/NAG with different values of c.
>
> We add additional experiments on GD and NAG with different values of $c$. The results are provided in Figure 5 in the PDF file in "Author Rebuttal". As illustrated, when $c$ is sufficiently large, increasing $c$ further has little effect on the convergence rate, which is consistent with our theory.
>
> [1] Zhang, H., Bi, Y., \& Lavaei, J. (2021). General low-rank matrix optimization: Geometric analysis and sharper bounds. Advances in Neural Information Processing Systems, 34, 27369-27380.
>
> [2] Bi, Y., Zhang, H., \& Lavaei, J. (2022, June). Local and global linear convergence of general low-rank matrix recovery problems. In Proceedings of the AAAI Conference on Artificial Intelligence (Vol. 36, No. 9, pp. 10129-10137).

---

> ### Author Response · Authors · 2024-08-12
>
> Dear Reviewer,
>
> Thank you for your time in reviewing our manuscript and offering valuable comments and suggestions. We've addressed the questions raised in your review with our author rebuttals. We'd like to know if these responses have sufficiently addressed your concerns. If any points require further clarification or discussion, please let us know.
>
> Thank you again for your effort.
>
> Best wishes,
>
> Authors

---

### Official Review · Reviewer_tURQ · 2024-07-09

**Soundness:** 2
**Presentation:** 2
**Contribution:** 2
**Rating:** 4
**Confidence:** 3

**Summary:**

This paper considers the convergence of the first-order optimization method, which includes the gradient descent and the nesterov's acccelerated gradient
for the marix factorization and the linear neural network.

In Section 2.1, they analyze the gradient descent algorithm on the matrix factorization (c.f. Thm. 1).
In Section 2.2, they analyze the NAG (c.f. Thm 2).  Section 3 illustrates the proof strategy.
Section 4 extends the analysis to the linear neural network (c.f. Thm. 3) and Section 5 presents the numerical experiments.

**Strengths:**

This paper is generally well-written and is quite easy to follow. Also, the analysis seems to be valid.

**Weaknesses:**

The major concern is the potential impact of this paper.  Generally speaking, this paper studies a well-studied problem with quite standard technique.

Also, there are some small typos that are quite annoying. For example, the $\|A\|_F$  in Thm. 1and Thm. 2 can cancel out. Thm 2 (line 14), the GD should be NAG. Please do another round of proof-reading.

**Questions:**

1. Why the convergence rate is independent of $\|A\|_F$?
2. Can you explain the line 807? Why the eigenvalues of $\lambda_i(T_{GD})$ ($1\leq i \leq (m-r)n$ can be ignored? The previous line suggest these eigenvalues are one.

---

> ### Author Rebuttal · Authors · 2024-08-07
>
> Thank you for your time in reviewing the paper and helping us improve.
> Our responses to each point of the Weaknesses and Questions are listed below.
>
> > W1: The major concern is the potential impact of this paper. Generally speaking, this paper studies a well-studied problem with quite standard technique.
>
> Impact is a little subjective, but we hope the reviewer could consider the following regarding whether this is a well-studied problem and whether our technique is standard:
>
> Quantitative theoretical guarantee for the optimization of general nonconvex function without Lipschitz gradient, even after decades of remarkable progress, is still largely an open problem. We managed to do it for a specific type of problem, namely matrix factorization. It is still a nonconvex problem without Lipschitz gradient. Moreover, it carries a lot of insight for a better understanding of deep learning, as it is closely related to linear neural networks, which we also analyzed. Therefore, it is extensively studied, however *not well-studied* because many understandings are still lacking.
>
> To illustrate this more precisely, consider [1] which is a very recent progress (NeurIPS'23): the motivation was to understand Gradient Descent (not gradient flow, which was easier) for this problem, but due to technical difficulty that was not accomplished; instead, the authors managed to work out a variant, namely Alternating Gradient Descent, and obtained convergence rate. Now, in our work, we not only work out Gradient Descent but also its momentum version, which is more complicated to analyze but beneficial, because we quantitatively prove that Nesterov's momentum indeed accelerates convergence. This requires *new analysis techniques* (as discussed in Remark 1) and rather precise (if not tight) error bounds, and we're glad it can work out.
>
> [1] Ward, Rachel, and Tamara Kolda. "Convergence of alternating gradient descent for matrix factorization." Advances in Neural Information Processing Systems 36 (2023): 22369-22382.
>
> > W2: There are some small typos that are quite annoying.
>
> Thank you very much for helping us catch them.
> Although we intentionally put one $|A|_F$ in the denominator of the prefactor so the other $|A|_F$ can align with the definition of relative error (lines 124 and 140), the expression indeed becomes more confusing.
> We will cancel out $|A|_F$ in Theorem 1 and 2 and $|LD^\top|$ in Theorem 3 to avoid confusion.
>
> The GD on line 140 is indeed a typo. We will proofread and fix it along with other typos in the next version.
>
> > Q1: Why the convergence rate is independent of $|A|_F$?
>
> While the convergence rate does not explicitly depend on $|A|_F$, $|A|_F$ will still affect the convergence rate through $c$ defined in Eq. (2). Theorems 1 and 2 require $c^2\geq\underline{c}^2=O(|A|_F)$, and the results show $|R_t|_F$ depends on $c^2$. When we choose $c=\underline{c}$, the bound will have an explicit dependence on $|A|_F$. When we choose $c>\underline{c}$, $|A|_F$ is dominated by the factor $c^2$, hence it is implicitly contained in our results in lines 123-144 and 139-140.
>
> Moreover, when considering the iteration complexity, we adopt relative error as the metric, i.e., $|R_t|_F/|A|_F\leq\epsilon$. Therefore, the right-most $|A|_F$ in the line between 123-124 (and the line between 139-140) does not appear in the iteration complexity.
>
> > Q2: Can you explain the line 807? Why the eigenvalues of $\lambda_i (T_{GD}) (1\leq i\leq (m-r)n)$ can be ignored?
>
> Thanks for the opportunity to correct a critical misunderstanding. It is *not* because eigenvalues are ignored, but because $\left<v,v_i\right>=0$. By Lemma 1 (line 188), $\mathcal{H}$ is the eigen subspace corresponding to positive eigenvalues of $H_0$, which is orthogonal to the kernel subspace of $H_0$. Through the derivation from line 804 to 807, we know that $\{v_1,\dots,v_{(m-r)n}\}$ exactly spans the kernel subspace of $H_0$ (where $\lambda_{mn-i}(H_0)=0$). Given the condition that $v\in\mathcal{H}$, we get $\left<v,v_i\right>=0$ for $i=1,\dots,(m-r)n$, hence the first $(m-r)n$ terms vanish.
> Thanks to the reviewer we realize this was under-explained, and will add an explanation about this in line 808 in the next version.

---

> ### Author Response · Authors · 2024-08-12
>
> Dear Reviewer,
>
> Thank you for your time in reviewing our manuscript and offering valuable comments. We've addressed the weaknesses and questions raised in your review with our author rebuttals. We'd like to know if these responses have sufficiently addressed your concerns. If any points require further clarification or discussion, please let us know.
>
> Thank you again for your effort.
>
> Best wishes,
>
> Authors

---

> ### Comment · Reviewer_tURQ · 2024-08-12
> **Thank you for the comment**
>
> Thank you for the response. I will stick to my original score.
>
> I appreciate that the author agrees with my suggestions on the presentation improvement and promise to make the revisions accordingly.

---

### Official Review · Reviewer_t3Co · 2024-07-13

**Soundness:** 3
**Presentation:** 3
**Contribution:** 3
**Rating:** 7
**Confidence:** 2

**Summary:**

The paper calculates convergence rates of the gradient descent and the Nesterov's accelerated gradient descent algorithms for factorization of rectangular matrices- a nonconvex optimization problem. Their analysis is for algorithms when the factor matrices are initialized as follows:  one matrix is initialized as the original matrix multiplied with Gaussian random matrix with appropriate scaling and the other is a zero matrix. They show convergence rates with quadratic dependence on condition number of the data matrix for GD and linear dependence for NAG. They also extend their analysis to linear neural networks and back their theoretical findings with empirical results.

**Strengths:**

1) Gradient descent algorithm and its variants are used almost everywhere in machine learning. Deep neural networks have made nonconvex optimization also very frequent in ML and matrix factorization is a basic problem. As such a work providing theoretical guarantees for the problem is very relevant.

2) Although the paper is difficult to read, it is structured in a very nice manner so that a reader can follow the high-level ideas quite clearly. The related works and the placement of this work among existing works is discussed very clearly.

3) I could not check the proofs in detail, but enough intuition, proof sketch and moderate level of details are provided in the main paper itself. The ideas appear solid and sound.

**Weaknesses:**

1) The experimentation is done on very small matrices. In practice much larger rectangular matrices are often encountered. It would give more insights if the experiments were also performed with moderate and large size matrices.

**Questions:**

See Weaknesses

**Limitations:**

See Weaknesses

---

> ### Author Rebuttal · Authors · 2024-08-07
>
> Thank you very much for your positive feedback and constructive comments.
>
> > W1: The experimentation is done on very small matrices. In practice much larger rectangular matrices are often encountered. It would give more insights if the experiments were also performed with moderate and large size matrices.
>
> Thank you for your suggestion. We add additional experiments on large-scale matrices. In particular, we set $(m,n)=(1200,1000)$ for matrix factorization and $(m,n,N)=(500,400,600)$ for linear neural networks in the additional experiment. We compare the performances of GD and NAG while keeping other settings the same as in Figure 2 in the paper. The results are plotted in Figure 4 in the PDF file attached to "Author Rebuttal". As illustrated in Figure 4, our conclusion in the paper that (1) NAG performs better than GD and (2) overparameterization accelerates convergence remains valid for large matrices.

---

> > ### Comment · Reviewer_t3Co · 2024-08-09
> >
> > I have read the other reviews and the rebuttal. I will keep my score.

---

### Official Review · Reviewer_SNTq · 2024-07-16

**Soundness:** 4
**Presentation:** 3
**Contribution:** 3
**Rating:** 7
**Confidence:** 4

**Summary:**

In this papers the authors analyze the convergence of  Nesterov Accelerated Gradient algorithm for a) rectangular matrix factorisation and b) linear neural networks. By using imbalanced initialisation,  the authors come up with linear rates of convergence impoving upon state-of-the-art regarding dependence of the rates condition numbers of sought matrices.

**Strengths:**

- The paper is well written and easy-to-follow.
- The analysis of convergence of NAG on rectangular matrix factorisation and linear neural networks is an interesting topic of research.
- The authors improved upon state-of-the-art results using novel technical approaches and elegant proof techniques.

**Weaknesses:**

- The authors didn't mention relevant works studying imbalance effect on the similar  problems e.g.  [1].
- The results address the linear neural network case but it's not obvious how the analysis could be extended to account for non-linearities or non-smooth activation functions.

[1] Min, Hancheng, et al. "On the explicit role of initialization on the convergence and implicit bias of overparametrized linear networks." International Conference on Machine Learning. PMLR, 2021.

**Questions:**

- In [1] the authors provided rates of convergence of gradient flows in the imbalance initialisation regime. Could these results provide some insights on how to generalise the derived rate for different amount imbalance of imbalance?
- Could the authors provide some insights on whether the theoretical results on linear neural network  could be extended to non-linear and possibly non-smooth activation functions?

Minor:
Eq. under line 132: Should $x_t$  be replaced by $z_t$ (??

**Limitations:**

Yes

---

> ### Author Rebuttal · Authors · 2024-08-07
>
> Thank you very much for your positive feedback and constructive comments. Our responses to each of the comments are listed below.
>
> > W1\& Q1: The authors didn't mention relevant works studying imbalance effect on the similar problems e.g. [1]. In [1] the authors provided rates of convergence of gradient flows in the imbalance initialisation regime. Could these results provide some insights on how to generalise the derived rate for different amount imbalance of imbalance?
>
> We thank the reviewer for pointing out the related work, and we will cite [1] and discuss it in the next version.
> In particular, our settings and proof techniques differ from [1]. Consequently, their results cannot directly translate into our case.
>
> 1. We consider GD and NAG with step size $O(1/L)$, while [1] considers GF with infinitesimal step size. Without carefully characterizing discretization error, the result in [1] for GF cannot be applied to GD.
>
> 2. In our proof, the imbalance initialization guarantees the induction steps to be valid. The amount of imbalance will affect the constant factors but will not affect the convergence rate $(1-\frac{\mu}{L})^t$ (or $(1-\sqrt{\frac{\mu}{L}})^t$). To be more explicit, suppose we initialize $X_0=c_1A\Phi_1\in\mathbb{R}^{m\times d}$, $Y_0=c_2\Phi_2\in\mathbb{R}^{n\times d}$, then by replacing $H_0$ in Proposition 2 with $H_0^\prime=I\otimes (X_0X_0^\top)$, we can generalize the proof in lines 814-815 and 821-832. The induction requires a sufficiently large $c_1$ and a relatively small $c_2\leq O(c_1)$. Meanwhile, by Proposition 4.3.2 in [2], we have $$\frac{1}{1-O(c_1c_2)}|A_0|_F\leq|R_0|_F\leq (1+O(c_1c_2))|A_0|_F$$ with high probability. Therefore, when $c_1$ is fixed, a smaller $c_2$ yields a smaller initial loss, resulting in a smaller constant factor. However, the convergence rate remains the same, as it depends on the extreme non-zero eigenvalues of $H_0^\prime$, i.e., $\mu$ and $L$, which only relates to $c_1$ but not $c_2$. We conduct additional experiments on GD/NAG with different values of $c_2$, and the numerical results support our claim. Please find the experiment details and results in "Author Rebuttal" and Figure 6 in the PDF file.
>
> > W2\& Q2: The results address the linear neural network case but it's not obvious how the analysis could be extended to account for non-linearities or non-smooth activation functions. Could the authors provide some insights on whether the theoretical results on linear neural network could be extended to non-linear and possibly non-smooth activation functions?
>
>
> Extending our results to neural networks with non-linear (and possibly non-smooth) activations is an interesting topic that we want to investigate in the future.
> While non-linearity complicates the analysis, we believe some of our techniques still apply.
> Suppose we apply non-linear activations $\sigma(\cdot)$ and the problem becomes $\min_{X,Y}|L-X\sigma(Y^\top D)|_F^2$. In our analysis, one of the key steps is to show that the residual and error terms are in the contraction subspace of the dynamics.
> Since the second layer is linear, by sketching initialization $X_0=cL\Phi$ we can still get $X_0$ that shares the same column space with $L$. By checking the dynamics of GD/NAG, we can verify that this space contains the column space of residuals and errors of all later iterations.
> However, due to the non-linear activation, the analysis of the linear part of the system and error terms becomes complicated.
> It requires further effort to verify whether our results can successfully generalize to the non-linear activation setting.
>
> > Q3: Eq. under line 132: Should $x_t$ be replaced by $z_t$?
>
> Thank you for pointing this out. This is indeed a typo and we will replace $x_t$ with $z_t$ in the next version.
>
> [1] Min, Hancheng, et al. "On the explicit role of initialization on the convergence and implicit bias of overparametrized linear networks." International Conference on Machine Learning. PMLR, 2021.
>
> [2] Ward, Rachel, and Tamara Kolda. "Convergence of alternating gradient descent for matrix factorization." Advances in Neural Information Processing Systems 36 (2023): 22369-22382.

---

> > ### Comment · Reviewer_SNTq · 2024-08-11
> >
> > Thank you for your rebuttal to my comments and for the additional experiments you conducted to explore how imbalanced initialization affects the rate of convergence. I will keep my score as it is.

---

### Author Rebuttal · Authors · 2024-08-07

We thank all the reviewers for dedicating their time to reviewing our paper and providing valuable feedback.
In response to the comments involving the size of the matrices (by reviewer t3Co), the performance of GD/NAG with different values of $c$ (by reviewer u7PK), and the amount of imbalance (by reviewer SNTq), we conduct additional numerical experiments and show the results (Figures 4 to 6) in the attached PDF file. We discuss each of them below:

> W1 by t3Co: The size of the matrices is small.

Our original submission uses $(m,n)=(100,80)$ for matrix factorization and $(m,n,N)=(100,80,120)$.
In this rebuttal where we investigate larger-sized problems, we set $(m,n)=(1200,1000)$ for matrix factorization and $(m,n,N)=(500,400,600)$ for linear neural networks. We compare the performances of GD and NAG under the same setting as in Figure 2 with moderate/large matrices in these sizes. The results are provided in Figure 4. As illustrated, the conclusion that (1) NAG performs better than GD and (2) overparameterization accelerates convergence remains valid for large matrices.

> Q8 by u7PK: The performance of GD/NAG with different values of $c$.

We conduct additional experiments on GD and NAG with different values of $c$ and plot the results in Figure 5.
As illustrated, when $c$ is sufficiently large, increasing $c$ further has little effect on the convergence rate, which is consistent with our theory.

> Q1 by SNTq: The effect of the amount of imbalance at initialization.

We conduct additional experiments on GD and NAG with initialization $X_0=c_1A\Phi_1\in\mathbb{R}^{m\times d}$, $Y_0=c_2\Phi_2\in\mathbb{R}^{n\times d}$, where $[\Phi_1]\_{i,j}\sim{N}(0,1/d)$ and $[\Phi_2]_{i,j}\sim{N}(0,1/n)$. We keep $c_1=50$ and set different values of $c_2$. The results are plotted in Figure 6. As illustrated, changing $c_2$ within a range will not significantly affect the convergence rate (slope), but will change the initial loss (intercept). This result supports our claim in the individual response to reviewer SNTq.

In the next version, we will add these results and discussions in an "Additional Experiments" section in the appendix.
For responses to other comments, please refer to our responses to reviewers.

---

### Decision · Program_Chairs · 2024-09-25

**Decision:**

Accept (poster)

**Comment:**

The paper studies the problem of factorization of rectangular matrices. The authors prove the convergence rate of gradient descent with quadratic dependence on the condition number of the data matrix, and the convergence rate of Nesterov's accelerated gradient descent with linear dependence on the condition number. They also extend their analysis to linear neural networks. Three reviewers are unanimously positive, and one review with borderline reject is short and less informative. I decide to accept the paper.